# Estimating global landslide susceptibility and its uncertainty through ensemble modeling

**Anne Felsberg[1], Jean Poesen[1,2], Michel Bechtold[1], Matthias Vanmaercke[1], and Gabriëlle J. M. De Lannoy[1]**

[1]Department of Earth and Environmental Sciences, KU Leuven, Heverlee, Belgium
[2]Faculty of Earth Sciences and Spatial Management, Maria-Curie Skłodowska University, Lublin, Poland

**Correspondence:** Anne Felsberg (anne.felsberg@kuleuven.be)

**Abstract.** This study assesses global landslide susceptibility (LSS) at the coarse 36 km spatial resolution of global satellite soil moisture observations to prepare for a subsequent combination of a global LSS map with dynamic satellite-based soil moisture estimates for landslide modeling. Global LSS estimation contains uncertainty, arising from errors in the underlying data, the spatial mismatch between landslide events and predictor information, and large-scale LSS model generalizations. For a reliable uncertainty assessment, this study combines methods from the landslide community with common practices in meteorological modeling to create an ensemble of global LSS maps. The predictive LSS models are obtained from a mixed effects logistic regression, associating hydrologically triggered landslide data from the Global Landslide Catalog (GLC) with predictor variables describing the landscape. The latter are taken from the Catchment land surface modeling system (including input parameters of soil (hydrological) properties and resulting climatological statistics of water budget estimates), as well as geomorphological and lithological data. Road network density is introduced as a random effect to mitigate potential landslide inventory bias. We use a blocked random cross validation to assess the *model uncertainty* that propagates into the LSS maps. To account for other uncertainty sources, such as *input uncertainty*, we also perturb the predictor variables and obtain an ensemble of LSS maps. The perturbations are optimized so that the *total predicted uncertainty* fits the observed discrepancy between the ensemble average LSS and the landslide presence or absence from the GLC. We find that the most reliable *total uncertainty* estimates are obtained through the inclusion of a topography-dependent perturbation between 15 % and 20 % to the predictor variables. The areas with the largest LSS un-

certainty coincide with moderate ensemble average LSS, because of the asymptotic nature of the LSS model. The spatial patterns of the average LSS agree well with previous global studies and yield areas under the receiver operating characteristic between 0.84 and 0.92 for independent regional to continental landslide inventories.

## 1 Introduction

Mitigating landslide impacts requires a good understanding of the spatial and temporal patterns of landslide occurrence. The spatial likelihood of a landslide is referred to as landslide susceptibility (LSS) and plays a crucial role in risk assessment and land use planning (Guzzetti et al., 2005; Crozier, 2013; Reichenbach et al., 2018). Regional high-resolution LSS maps derived from environmental conditions are a fundamental tool for informing local population, city planners, and decision makers both on the immanent landslide likelihood but also about secondary effects such as major sediment sources (Crozier, 2013; Maes et al., 2017; Broeckx et al., 2020). Large-scale low-resolution LSS maps can serve as background information to be downscaled for the above applications at the local scale, or they can be used in conjunction with large-scale satellite data to construct a spatiotemporal estimate of the likelihood for a landslide.

Due to their generalizing nature, LSS models are prone to uncertainty (Petschko et al., 2014). A large number of LSS models exists, but most focus on local to regional scales and typically lack thorough validation or uncertainty assessment (Reichenbach et al., 2018). Recent advances in computational power and data availability have fostered the develop-

ment of LSS maps at continental level (for example Europe, Wilde et al., 2018, and Van Den Eeckhaut et al., 2012; and Africa, Broeckx et al., 2018) or at the global scale (for example Nadim et al., 2006; Hong et al., 2007; Lin et al., 2017; Stanley and Kirschbaum, 2017). While information about the uncertainty would be essential to know how reliable these large-scale LSS maps are as well as how much variation can be expected within a mapping unit, only Broeckx et al. (2018) provide such a measure for their map of Africa and only to a limited degree. The quantification of LSS uncertainty becomes even more called for yet challenging at the global scale and with coarser spatial resolution due to necessary generalizations and the increased spatial mismatch between landslide events and predictor information. A reliable uncertainty assessment of global LSS estimates is moreover crucial when subsequently combining them in a statistically optimal way with, for example, satellite soil moisture products from Soil Moisture and Ocean Salinity (SMOS) or Soil Moisture Active Passive (SMAP) as used by Felsberg et al. (2021).

Uncertainty is typically grouped according to its origin into *model uncertainty* (here "how correct are the equations that we use to predict LSS?") and *input uncertainty* (here "how correct is the input to these equations?"). Model uncertainty stems from heuristic choices that are necessary in the process of model creation, including the choice of the statistical modeling approach, the selection of predictor variables, training data sampling and training data quality (see for example Steger et al., 2015; Pourghasemi and Rossi, 2016; Zêzere et al., 2017; Depicker et al., 2020; Lima et al., 2021). In order to estimate some of these model-intrinsic errors for a chosen modeling approach, cross validation (CV) is a widely used method where data are divided into a number of subsets, which are subsequently used for training and testing of the model. How to best sample the CV subsets to retrieve realistic uncertainty estimates is in itself a field of research. For LSS maps, random sampling is most common (see for example Broeckx et al., 2018), while spatial sampling is used less often for an additional uncertainty estimate (see for example Steger et al., 2020, or Depicker et al., 2020). However, these are known to respectively strongly underestimate and possibly overestimate the model uncertainty, and hybrid methods such as blocked random CV (B-CV) have been suggested to result in the most reliable uncertainty estimates (Roberts et al., 2017). CV leads to multiple LSS model equations (one per CV subset), and the standard deviation of the resulting LSS values gives an indication of the associated model uncertainty as shown by Broeckx et al. (2018) for Africa.

Input uncertainty principally results from errors in the environmental data. To assess how input uncertainty propagates into the total predicted uncertainty, ensemble simulations can be used. Meteorologists, for example, simulate the weather based on a distribution of initial conditions and predict an ensemble of equally possible outcomes (ensemble members). Instead of only one deterministic weather fore-

cast, they use the ensemble average prediction that has been found to perform better than their deterministic counterpart (Kalnay et al., 2006). The uncertainty of the final ensemble average prediction can then be estimated by the variance or standard deviation among the ensemble members.

The *total ensemble uncertainty*, resulting from the combination of these methods that account for model and input uncertainty respectively, is assumed to be reliable if it matches the observed "actual" *total uncertainty*. The latter is estimated by comparing the predicted average LSS against the observed presence and absence of landslides. The gap between this observed and the predicted total uncertainty can then be closed by tuning the magnitude of the ensemble input perturbations. Note that this implies that the perturbations might in the end not purely capture the input uncertainty but actually compensate for other sources of uncertainty as well that are not specifically addressed. One such important source of uncertainty is spatial representativeness error (Blöschl and Sivapalan, 1995; van Leeuwen, 2015), especially when evaluating spatially averaged grid cell LSS estimates using single landslide observations as reference data.

In this study, we combine CV and an ensemble approach to create global LSS maps with a reliable total uncertainty (full ensemble standard deviation). We create multiple LSS equations as part of CV (weak model constraint) and subsequently perturb the selected predictor variables (input of the LSS model equations) to retrieve a full ensemble of possible LSS values. Specifically, we focus on hydrologically triggered landslides and propose to include long-term climatological statistics of hydrometeorological variables as predictor variables, in addition to the common geomorphological ones. We use a mixed effects logistic regression (MELR) relying on the strong generalizing capabilities of logistic regression as the basic model structure, and we mitigate the potential reporting bias of landslide presences in the GLC with stratified average road network density (RND) as a random effect. To limit biases from unreliable and confounding definitions of landslide absence grid cells for the model creation, we introduce a novel approach based on a characteristic distance between landslides. After having taken these steps to limit the introduced uncertainty, the B-CV is used to instill model uncertainty via a selection of different possible predictor variables and associated parameters, and we further add (and tune) ensemble perturbations to the selected predictor variables to obtain a reliable total ensemble uncertainty. This LSS assessment is carried out on the 36 km Equal-Area Scalable Earth version 2 (EASEv2) grid, in line with the nominal spatial resolution of satellite soil moisture estimates from SMOS or SMAP. Producing spatial LSS estimates at this resolution facilitates a subsequent combination with the satellite-based temporally dynamic data, as well as calculations of the above-mentioned climatological statistics and the development of computationally intense ensemble approaches. To our knowledge, no framework has previously

been developed for the assessment of the total uncertainty of LSS predictions.

Section 2 introduces the landslide (presence, absence) and environmental data used to create ensemble LSS maps. The LSS model construction based on MELR is introduced in Sect. 3, along with the methods of CV and input predictor variable perturbations for uncertainty assessment, as well as methods to evaluate the results. Section 4 presents the resulting LSS model structure and selected predictor variables, as well as the ensemble LSS evaluation for different input perturbations. Section 5 discusses various aspects of the results. The paper closes with a summarizing conclusion.

## 2 Data

### 2.1 Landslide data

A first step in creating our LSS models is the creation of suitable training datasets, indicated in the upper part of the flowchart in Fig. 1. We use reported hydrologically triggered landslide occurrences from the most recent version of the GLC (https://landslides.nasa.gov/viewer, last access: 8 February 2021). The GLC is a landslide inventory that contains information about location, date and trigger. It is originally based on media reports (Kirschbaum et al., 2010, 2015) but has recently been supplemented with the citizen-science-based Landslide Reporter Catalog (LRC) data (Juang et al., 2019); see Stanley et al. (2021) for details. Any reference to the GLC hereafter refers to this combined data product. Despite known English-language and economic biases (Kirschbaum et al., 2010, 2015), the GLC covers all continents and landslide hotspots. It has already been used for the creation of two global LSS maps (Stanley and Kirschbaum, 2017; Lin et al., 2017) and was used to train the newest version of the Landslide Hazard Assessment for Situational Awareness (LHASA) model version 2.0 (Stanley et al., 2021).

For this study, we use 12515 hydrologically triggered landslides (GLC classifiers "continuous rain", "downpour", "monsoon", "flooding", "rain" and "tropical cyclone") reported mainly between January 2007 and November 2020. Since LSS informs about the static environmental landslide likelihood, it is common practice to exclude the temporal aspect of landslide occurrence and instead work with landslide presence and absence locations. Multiple landslides within the same 36 km EASEv2 grid cell are therefore aggregated into one landslide presence grid cell, resulting in a total of $N_{LS} = 3757$ (orange grid cells, Fig. A1). While we acknowledge that grid cells with more frequent landslide reporting can in general be expected to have a higher LSS, we found that the information about the frequency of landslide occurrence within a grid cell strongly mirrors biases in the landslide inventory; e.g., more landslides are reported in English-speaking countries. The aggregation, on the contrary, reduces

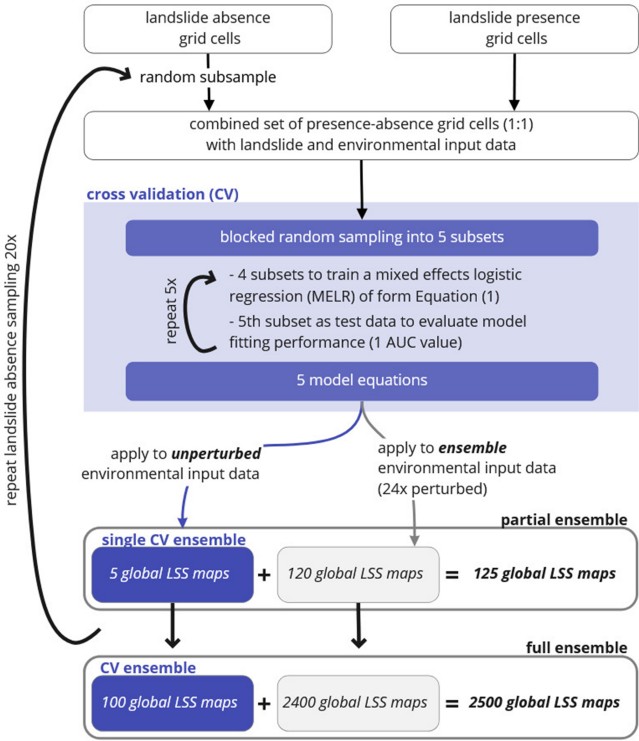

**Figure 1.** Schematic of methodology used in this study to derive ensembles of global landslide susceptibility (LSS) maps. "Ensemble" refers to a collection of LSS maps. In the course of this study, we refer to different subsets of the full ensemble ($LSS_{2500}$), namely the ensemble from one single blocked random CV application (single CV ensemble, $LSS_5$), when adding input perturbations to it (partial ensemble, $LSS_{125}$) or when repeating the underlying landslide absence subsampling (CV ensemble, $LSS_{100}$). Subscript numbers indicate the size of the LSS ensemble. Model fitting performance is evaluated during the process of cross validation (CV) by calculating the area under the receiver operating characteristic curve (AUC) for each model equation of form Eq. (1).

the landslide presence reporting bias of the GLC. To address the remaining landslide presence bias originating from more landslide reporting in frequently accessed areas, we use stratified data on the RND (including highways and all types of roads, ranging from primary to local roads) provided by the Global Roads Inventory Project (GRIP) (Meijer et al., 2018) as a random effect, explained in Sect. 3.1.

The creation of realistic statistical LSS models and uncertainty estimates depends on the knowledge of both landslide presences and absences (Roberts et al., 2017; Steger and Glade, 2017; Knevels et al., 2020; Lucchese et al., 2021). Usually, an absence grid cell is simply defined as one without a recorded landslide. For local modeling, this might work when complete and reliable landslide inventories are available. For large or remote areas, however, no reported landslide does not necessarily mean that the site never experienced one. Terrain features show a certain amount of spa-

tial autocorrelation indicating that locations in proximity of a known landslide are generally prone to instability as well. The use of grid cells too close to known landslide locations as an absence reference should therefore be avoided (Brenning, 2005). On the other hand, absence grid cells sampled very far from the reported landslide locations, in so-called "trivial" or easily classifiable areas (for example flat areas), might result in an underrepresentation of stable areas in the vicinity of the known landslide locations (Steger and Glade, 2017). Additionally, it might confound the selection process of geomorphologically meaningful predictor variables and lead to an overoptimistic conception of the resulting LSS map's quality (Steger and Glade, 2017; Lucchese et al., 2021).

In this study, we therefore adopt a sampling strategy as used in earlier LSS assessments (Van Den Eeckhaut et al., 2012; Lin et al., 2017; Zhu et al., 2017; Nowicki Jessee et al., 2018; Knevels et al., 2020; Lucchese et al., 2021), where reliable absence grid cells are defined between a minimum (buffer) and maximum radius around known landslide presence grid cells. As a measure of spatial autocorrelation we derive the characteristic distance between two landslides from the GLC (for details see Appendix A1). We use this characteristic distance of 221.43 km ($\sim 6$ grid cells) as the buffer radius, and we use 2.5 times this distance ($\sim 15$ grid cells) as maximum radius. Absence grid cells are hence selected from grid cells 7 to 15 around a landslide occurrence (blue grid cells in Fig. A1). This definition still results in more than 6 times more absence grid cells ($N_{\text{noLS}} > 25\,000$) than landslide presence grid cells ($N_{\text{LS}} = 3757$). We therefore randomly sample from the absence grid cells with a $1 : 1$ ratio ($N_{\text{LS}} : N_{\text{noLS}}$) as is commonly done, for example by Brenning (2005), Steger and Glade (2017), Nowicki Jessee et al. (2018), Depicker et al. (2020), Knevels et al. (2020), Lin et al. (2021) and Lucchese et al. (2021). LSS models are subsequently constructed based on data from 7514 (absence + presence) grid cells, as illustrated in Fig. 1.

## 2.2 Environmental data

The 77 predictor variables considered in this study are listed in Table 1 and were selected based on earlier reviews on the most common predictors used for LSS maps (Pourghasemi and Rossi, 2016; Reichenbach et al., 2018). In statistical LSS models, these predictor variables act as proxies for one or multiple processes underlying a landslide (Whiteley et al., 2019). Since LSS is referring to the spatial likelihood of landslides, we only consider predictor variables that are (quasi-)static in time.

To better represent processes underlying hydrologically triggered landslides, we include long-term climatological statistics of soil moisture in different layers, soil surface temperature, runoff, rainfall, evaporation and snow depth as possible predictor variables. These climatological statistics include the range (here defined as the difference between percentiles 1 and 99), inter-quartile range, mean, median, per-

centile 99 and maximum within the time period 1990–2020, derived from 36 km simulations with the CLSM (Koster et al., 2000; Reichle et al., 2019), forced with Modern-Era Retrospective analysis for Research and Applications, Version 2 (MERRA-2) meteorological data, as in Felsberg et al. (2021).

Most other predictor variables are part of the 36 km input parameters to the CLSM. Of these, elevation and compound topographic index (CTI) stem from the same underlying Shuttle Radar Topography Mission (SRTM) data as the morphological information on slope from the United States Geological Survey (USGS), but with different data sources for the high northern latitudes (Verdin et al., 2007).

We use lithological information from the Global Lithological Map (GLiM) (Hartmann and Moosdorf, 2012) aggregated to the fraction of a grid cell covered by each of the 13 lithological classes (we exclude the classes "water", "ice and glacier", and "no data"). This produces a dataset with 13 fields, each with a continuous fraction estimate. Peak ground acceleration (PGA) is the likely level of ground motion from earthquakes (Giardini et al., 2003). Here, we do not use it as the likelihood of a seismic landslide trigger but rather as a proxy for the fracturing and weakening that lithologies have undergone due to seismic and tectonic activity (Lin et al., 2017; Vanmaercke et al., 2017; Broeckx et al., 2018). Details on the aggregation methods are given in Table 1.

## 3 Model construction and evaluation

This section introduces the methods used in this study for model construction and evaluation. Section 3.1 introduces the general principles of logistic regression used to derive global LSS estimates, before elaborating the predictor variable selection process and the implementation of average road network density as a random effect. Section 3.2 introduces methods for uncertainty assessment. First, cross validation is introduced with a detailed explanation of the blocked random sampling. Second, the methods of input ensemble perturbations are briefly explained (details are elaborated in Appendix A2). LSS results based on the first approach alone are referred to as "CV ensemble" or $\text{LSS}_{100}$. Results based on both CV and input ensemble perturbations are referred to as "full ensemble" or $\text{LSS}_{2500}$. Section 3.3 introduces the methods and data used for the evaluation of ensemble average LSS and the impact of the extended uncertainty assessment through input perturbations.

### 3.1 Mixed effects logistic regression (MELR) for model development

In this study, we create a statistical LSS model using MELR (Zuur, 2009), as previously also employed by Steger et al. (2017), Lin et al. (2021) and Lima et al. (2021). Logistic regression is the most commonly used approach for statis-

**Table 1.** Environmental predictor variables used in this study, alongside their data source, original spatial resolution and methods used for aggregation to the 36 km EASEv2 grid. Apart from slope, lithology, PGA and rainfall, the specified aggregation was not conducted in this study. Predictor variables that are part of the CLSM parameter set or output do not require any spatial aggregation. Long-term climatological statistics of all hydrological variables comprise the range (here the difference between 1st and 99th percentile), inter-quartile range, mean, median, 99th percentile, and maximum between 1990 and 2020. MERRA-2 precipitation is used as input for the calculations of the hydrological climatological statistics and has been interpolated to the 36 km EASEv2 grid as part of the simulation process. Units are given for the original data but are removed through the rescaling of the data to the interval (0, 1) (see text).

| Predictor variables | Data source | Original spatial resolution | Aggregation method to or within EASEv2, 36 km grid cell |
|---|---|---|---|
| slope (mean, maximum) [°] | USGS: details in Verdin et al. (2007) based on SRTM DEM[a] and GTOPO30[b] | $3''$ (SRTM DEM), $30''$ (GTOPO30) | mean and maximum |
| elevation (mean, standard deviation) [m a.s.l.] | CLSM parameters: details in Verdin (2013) based on SRTM DEM[a] and GMTED2010[c] | $3''$ (SRTM DEM), $7.5''$ (GMTED2010) | mean and standard deviation |
| depth to bedrock [m] | CLSM parameters: details in De Lannoy et al. (2014) based on GSWP-2[d] | $1°$ | spatial interpolation |
| percentage of gravel (0–30 cm) [vol %] | CLSM parameters: | $30''$ | most representative |
| percentage of clay (0–30 and 0–100 cm) [w %] | details in De Lannoy et al. (2014) | | $30''$ sample |
| percentage of sand (0–30 and 0–100 cm) [w %] | based on STATSGO2[e] | | |
| porosity (0–30 and 0–100 cm) [m³ m⁻³] | and HWSD1.21[f] | | |
| wilting point divided by porosity (0–30 and 0–100 cm) [–] | | | |
| compound topographic index, CTI (mean, maximum) = ln(specific catchment area/tan(slope)) [log(m)] | CLSM parameters: details in Verdin (2013) based on SRTM DEM[a] and GMTED2010[c] | $3''$ (SRTM DEM), $7.5''$ (GMTED2010) | mean and maximum |
| land fraction within grid cell [–] | CLSM parameters: HYDRO1k based on GTOPO30, 1996 (EROS, 2018; Verdin, 2013) | $10''$ | areal fraction |
| fraction covered by each of 13 lithological classes [–]: metamorphic rocks, mixed sedimentary rocks, siliclastic sedimentary rocks, basic plutonic rocks, acid plutonic rocks, basic volcanic rocks, intermediate volcanic rocks, carbonate sedimentary rocks, unconsolidated sediments, intermediate plutonic rocks, pyroclastics, evaporites, acid volcanic rocks | GLiM created by Hartmann and Moosdorf (2012) | polygons | areal fraction |
| peak ground acceleration, PGA [m s⁻²] due to earthquakes expected with a return period of 475 years (i.e., 10 % exceedance probability in 50 years) | GSHM[g] created by GSHAP[h] (Giardini et al., 2003) | $1°$ | nearest neighbor |
| rainfall climatological statistics [mm] | MERRA-2 (Bosilovich et al., 2016) | $0.625°$ long $\times$ $0.5°$ lat | bilinear interpolation |
| surface soil moisture climatological statistics (0–5 cm) [m³ m⁻³] | CLSM output | EASEv2, 36 km | – |
| root zone soil moisture climatological statistics (0–100 cm) [m³ m⁻³] | | | |
| profile soil moisture climatological statistics (0–100 cm) [m³ m⁻³] | | | |
| land surface temperature climatological statistics [K] | | | |
| runoff climatological statistics [mm] | | | |
| evaporation (incl. transpiration) climatological statistics [mm] | | | |
| snow depth climatological statistics [mm] | | | |

[a] Shuttle Radar Topography Mission digital elevation model. [b] USGS global elevation model. [c] Global Multi-resolution Terrain Elevation Data 2010. [d] Second Global Soil Wetness Project. [e] U.S. General Soil Map. [f] Harmonized World Soil Database version 1.21. [g] Global Seismic Hazard Map. [h] Global Seismic Hazard Assessment Project.

tical LSS mapping (Reichenbach et al., 2018). It is associated with strong generalizing capabilities (Brenning, 2005), which is a necessity when working at the global scale, and it has proven to be reliable in continental to global LSS assessments (Broeckx et al., 2018; Lin et al., 2017). Within logistic regression, the LSS, here defined as the probability of a landslide presence within a grid cell, $P(Y = 1)$, is given by

$$P(Y = 1) = \frac{\exp(\alpha + \sum_{i=1}^{n} \beta_i x_i)}{1 + \exp(\alpha + \sum_{i=1}^{n} \beta_i x_i)}, \quad (1)$$

with $\alpha$ [–] the intercept, $x_i$ [–] the independent predictor variables, $\beta_i$ [–] the associated coefficient and $n$ the number of predictor variables. A one unit change in the predictor variable $x_i$ results in a multiplicative change by $\exp(\beta_i)$ in the odds of landslide presence, defined as the ratio of $P(Y = 1)/(1 - P(Y = 1)) = \exp(\alpha + \sum_{i=1}^{n} \beta_i x_i)$. An increase in the odds of landslide presence is associated with a (nonlinear) increase in LSS. Positive (negative) $\beta$ values hence indicate an increase (decrease) in LSS with an increase in the predictor variable. In this study, we work with rescaled predictor variables (between their global minimum and maximum) to detach the magnitude of the $\beta$ values from the magnitude of the predictor variable. This facilitates subsequent interpretation.

We employ a stepwise forward technique to select five predictor variables, corresponding to the commonly used number of predictor variables for LSS assessment at the global scale (Nadim et al., 2006; Stanley and Kirschbaum, 2017; Lin et al., 2017; Reichenbach et al., 2018). Based on the Akaike information criterion (AIC), a measure that is proportional to the sum of squared errors and allows for comparison between non-nested models, we determine the best-performing univariate MELR, i.e., the first predictor variable. The AIC comparison is subsequently repeated for multivariate MELR with one additional predictor variable at a time. This stepwise forward selection also allows us to exclude correlated predictor variables ($r > 0.7$, following for example Dormann et al., 2013), so that largely independent predictor variables are used in the logistic regression. An analysis of the generated models using the variance inflation factor (VIF) proved that this approach indeed successfully prevented a logistic regression model construction based on predictor variables that are too strongly correlated.

The mixed effects approach allows us to include a categorically scaled variable as a so-called "random effect", here the random intercept $\alpha$, for which we use the average road network density (RND) stratified into six classes. We summarize all land grid cells where average RND is negligible ($< 1\,\mathrm{m\,km^{-2}}$) into the first class and use quantiles 20, 40, 60 and 80 of those grid cells with non-negligible RND to divide the rest into additional five classes. The mixed effects approach will then result in one global logistic regression equation that has the same $\beta$ factors for all grid cells but six different $\alpha$ values according to each grid cell's RND class. For model fitting purposes it is assumed that these six $\alpha$ values come from a normal distribution (Zuur, 2009).

The underlying assumption of RND as a random effect is that the representativeness of the landslide data from the GLC varies with the RND of the region. We recognize that RND may also serve as a proxy for human interference or likelihood of slope cutting and may hence be included as a predictor variable, as was argued by Stanley and Kirschbaum (2017). The use of RND as a predictor variable or random effect can be expected to have similar results if the connected bias were small. For large biases, however, predictions using RND as a predictor variable would systematically underestimate the actual LSS of remote areas with strong underreporting of landslides (as was put forward by Steger et al., 2017, for forested areas). The inclusion of RND as a random effect favors the selection of natural, physically valid predictor variables while allowing for locations without roads to also receive a high predicted LSS. The inclusion of random effects in a regression model results in unbiased model parameter estimates, but it does not inform about the uncertainty of the predictions (Roberts et al., 2017). We use the glmer function from the lme4 package (Bates et al., 2015) to create MELR models in R version 4.0.3 (R Core Team, 2020) where the best-fitting parameters are obtained by maximum likelihood estimation.

### 3.2 Cross validation (CV) and input perturbations for reliable uncertainty estimation

In this study, the predicted total ensemble uncertainty results from the combination of CV techniques and input ensemble perturbations. For CV, the data are separated into five subsets, which subsequently are used for training and testing the model with the hold-one-out technique, as illustrated in Fig. 1. We employ a blocked random CV (B-CV), as recommended by Roberts et al. (2017), which we found to indeed yield most realistic error estimates in comparison to random or spatial sampling (not shown). This means that instead of randomly sampling individual grid cells into the five subsets for training and testing the model as part of CV, we randomly sample small groups of grid cells with similar environmental conditions, so-called "blocks" (see Fig. 1). We expect that the environmental conditions are similar in neighboring pixels (for example same subcontinent) and for similar climate zones. We therefore derive blocks in two steps. First, the 7514 grid cells selected for model creation are divided according to 10 predefined (sub-)continents. Within each (sub-)continent, we then derive in a second step 10 blocks through $k$-means clustering (Lloyd, 1982) of 30-year average soil surface temperature and rainfall (see Table 1). In total we retrieve 100 blocks comprising different numbers of grid cells (median: 55) that are not necessarily located next to each other. The 100 blocks are then randomly divided into the five subsets for model creation (20 each).

Next, the MELR is iteratively trained on four subsets, and the model fitting performance is tested against the fifth, i.e., the hold-one-out subset, using each subset as a test subset once (see Fig. 1). This results in five different model equations of form Eq. (1) and corresponding LSS maps. By repeating the random absence grid cell subsampling 20 times, we obtain a total of 100 LSS maps (referred to as CV ensemble or $LSS_{100}$; see Fig. 1) that allow for calculations of an ensemble average LSS ($\overline{LSS}_{100}$), as well as a standard deviation ($\sigma_{LSS_{100}}$) per grid cell. Note that the definition of the individual blocks varies between each repetition of absence grid cell sampling due to the $k$-means clustering algorithm.

For the input ensemble perturbations, we apply one fitted model equation to a slightly perturbed set of its predictor variable values. In total, 24 repetitions of this process are conducted, resulting in a total ensemble of 25 LSS maps per model equation (see Fig. 1). In combination with the five model equations and 20 repetitions for the CV ensemble, this results in a total amount of 2500 LSS maps (referred to as full ensemble or $LSS_{2500}$) with corresponding average ($\overline{LSS}_{2500}$) and standard deviation ($\sigma_{LSS_{2500}}$) per grid cell. The latter is representative of the total predicted uncertainty.

The aim is to design an LSS model setup so that the predicted total ensemble uncertainty quantified by the ensemble variance or spread $\sigma_{LSS}^2$ matches the discrepancy between predictions and observations, which we refer to as the actual uncertainty. A measure of this actual uncertainty is the Brier score (BS) (Wilks, 2011), which compares the predicted average LSS ($\overline{LSS}$) against landslide observations from the GLC ($o$) at different grid cells $i$ ($i = 1, \ldots, N$):

$$BS = \frac{1}{N} \sum_{i=1}^{N} (\overline{LSS} - o)_i^2, \tag{2}$$

with $o$ being 1 for landslide presence and 0 for absence grid cells. This actual uncertainty by design includes model and input error ($\overline{LSS}$), but also error in the reference data ($o$), and spatial representativeness error. The perturbations to the predictor variables are randomly sampled from a normal distribution with the mean being the original value of the grid cell. The standard deviation, or perturbation magnitude, is tuned, so that the resulting total ensemble spread (including the spread originating from CV) matches the observed actual uncertainty BS in Eq. (2). For details of the tuning process, see Appendix A2. We apply the same perturbation magnitude to all (rescaled) predictor variables. The magnitude is chosen to increase proportionally to the topographic complexity of a location from 15 % to 20 %. We use the variation of elevation within a grid cell as a measure of said topographic complexity and find this perturbation scaling to yield better results than a globally constant perturbation magnitude. Note that these perturbations in $x_i$ do not linearly propagate into the LSS estimates, because the logistic regression (see Eq. 1) relates $x_i$ to LSS via an S-shape LSS curve, with quasi-linear behavior at the center (i.e., intermediate $x_i$ values) and asymptotic behavior towards the upper or lower limit (i.e., for very low or high $x_i$ values). Locations of largest perturbation do thus not necessarily coincide with large resulting ensemble uncertainty.

### 3.3 Evaluation

To quantify how well a predicted LSS map represents observed landslide presences and absences, a BS can be used (see Eq. 2). Alternatively, the receiver operating characteristic (ROC) is commonly used as evaluation tool for categorical response values such as landslide presence and absence (Reichenbach et al., 2018). For the ROC, the true positive rate of one LSS map is displayed against its false positive rate for different possible thresholds in the continuous probability (here LSS) that is predicted. The true positive rate is the proportion of correctly predicted landslide presence grid cells when applying said threshold (true positives) of all observed landslide presence grid cells (Wilks, 2011). The false positive rate is the proportion of erroneously predicted landslide presence grid cells (false positives) of all observed landslide absence grid cells. The area under the ROC curve (AUC) is 1 for a perfect representation of the spatial LSS distribution, whereas an AUC value of 0.5 indicates that the model does not perform better than a uniform distribution.

Depending on the reference landslide data, the ROC analysis can be conducted for specific grid cells from a CV subset (independent data not used in the training) or from other independent landslide inventories. Here, we use landslide presence and absence information from the grid cells of the fifth CV subset to assess the model fitting performance for each LSS ensemble member map on the go. To evaluate the final prediction performance of the complete ensemble averages and the corresponding ensemble members, we use three independent landslide inventories. We obtain 36 km landslide presence grid cells as described for the GLC in Sect. 2.1 for (i) quarterly reports issued by the Russian Federation (FSBIH, 2018) with $N_{LS} = 56$ aggregated from 183 observations; (ii) an inventory for Africa by Broeckx et al. (2018) with $N_{LS} = 649$ aggregated from 18 053 observations; and (iii) FraneItalia, a catalog of recent landslides in Italy (Calvello and Pecoraro, 2020) with $N_{LS} = 309$ aggregated from 5438 observations. Since we trust their landslide absence reporting to be reliable, we use all other grid cells within the region in question as landslide absence grid cells. These validation inventories cover different climatic zones and hence landslide regimes, stem from (mostly) non-English-speaking regions (Africa, Russia, Italy), and include less populated areas (Africa, Russia), which are not well represented in the GLC data that underlie our LSS estimates. With Italy being a hotspot of landslide occurrence within Europe, we are moreover able to assess whether the coarse spatial resolution hinders realistic regional assessment within smaller, potentially very susceptible areas.

The AUC and BS metrics can be computed for individual ensemble members (of the CV ensemble $LSS_{100}$ or the full ensemble $LSS_{2500}$, yielding a distribution of metrics) or for ensemble averages ($\overline{LSS}_{100}$ and $\overline{LSS}_{2500}$). It will be assessed whether (i) an ensemble average outperforms an individual member LSS realization and whether (ii) the full ensemble average with ensemble input perturbations ($\overline{LSS}_{2500}$) outperforms the CV ensemble average which does not include input perturbations ($\overline{LSS}_{100}$). This would be in line with the expectations for hydrological or meteorological models (Kalnay et al., 2006).

## 4 Results

### 4.1 LSS model structure

This section investigates the different values for the $\beta$ coefficients and intercept $\alpha$ of the 100 MELR models created following Fig. 1. The landslide absence data, used to train these models, differ for each of the 20 repetitions, and subsequently the definitions of the subsets for B-CV vary as well. All 100 models result in LSS maps with very high AUC values above 0.8, with a median of 0.92, for the corresponding test data.

The values of the intercept $\alpha$ take negative values for low RND and positive values for high RND (by design, not shown). The left panel of Fig. 2 shows which predictor variables were selected how often and during which step of the selection process (AIC; see Sect. 3.1). The right panel shows boxplots of the $\beta$ values for each predictor variable (see Eq. 1). Whiskers extend from minimum to maximum and boxes from 25th to 75th quantile, with the median indicated in between. The first selected predictor variable was always related to the slope, i.e., either the mean CTI within the grid cell, the maximum slope or the mean slope. The mean CTI, also known as a topographic wetness index, was selected as part of all 100 models. It is inversely proportional to slope (see Table 1), which is in line with the negative $\beta$ values, i.e., decrease in LSS is expected with increasing CTI. The second selected predictor variable is either another slope measure (maximum slope or standard deviation of the elevation, i.e., local relief) or, for more than 65 % of the models, related to the climatologic conditions (median surface soil moisture, range of evaporation, maximum evaporation or surface soil moisture). Out of these variables, median surface soil moisture stands out as most frequently being the second predictor variable (for more than 50 % of the models). Independent of the selection step, it is part of more than 80 % of the models. All of these variables are modeled with positive $\beta$ values; i.e., the higher the predictor variable, the larger the odds of a landslide presence and hence the LSS.

The areal fraction of evaporites within the grid cell is the only lithological class that was selected and only in the final selection step. The very unrealistic $\beta$ value associated with this predictor ($-128.65$) suggests that this selection is possibly a statistical artifact. The PGA, treated as a proxy for lithologic weakening due to regular seismic activity, is dominantly selected in the later variable selection steps but still part of about 80 % of the models.

### 4.2 Evaluation of ensemble LSS

Based on these 100 model equations and when perturbing the input parameters (see Fig. 1), we obtain the full ensemble average LSS ($\overline{LSS}_{2500}$) and standard deviation ($\sigma_{LSS_{2500}}$) shown in Fig. 3. The highest $\overline{LSS}_{2500}$ can be found in the large mountain ranges on all continents as well as coastal areas (especially the islands in Southeast Asia). Very flat areas or planes, such as central northern Canada, Siberia, the Tibetan Plateau, the Arabian Peninsula, large parts of Africa (especially the Sahara), and central Australia, have very low $\overline{LSS}_{2500}$. Intermediate $\overline{LSS}_{2500}$ values are found in the northern Rocky Mountains towards Alaska as well as the Kolyma Range in Russia, at the northeastern shores of South America and the western shores of Africa, along the East African Rift, Scandinavia and India. Figure 4a shows a density scatter plot of $\sigma_{LSS_{2500}}$ versus $\overline{LSS}_{2500}$. The uncertainty $\sigma_{LSS_{2500}}$ is large for areas with intermediate $\overline{LSS}_{2500}$, whereas very high or low $\overline{LSS}_{2500}$ typically have a smaller associated $\sigma_{LSS_{2500}}$.

Figure 5 illustrates the ensemble $LSS_{2500}$ distribution for 20 randomly sampled landslide presence and absence grid cells. Even though we quantify the uncertainty with a $\sigma_{LSS_{2500}}$, the distributions are mostly non-Gaussian. Most displayed landslide presence (absence) grid cells have LSS distributions ranging at the upper (lower) end of the interval (0,1). Grid cells 1, 7 and 18, however, exhibit a very wide distribution that seems disconnected from the absence (1, 17) or presence (18) of a landslide.

The ROC curves for ensemble average $\overline{LSS}_{2500}$ are shown in Fig. 6, with the curves for Russia (AUC: 0.92) and Italy (AUC: 0.91) being closest to the upper left corner and that for Africa being a little further from this optimum (AUC: 0.84). The $\overline{LSS}_{2500}$ map hence very well captures the landslide patterns over all three regions.

### 4.3 Impact of input perturbations

The above discussion of the full ensemble $LSS_{2500}$ includes perturbations to the predictor variables on top of the CV ensemble $LSS_{100}$ obtained by the CV techniques alone. Figure 4a and b show that the LSS uncertainty is a function of the average LSS values and that $\sigma_{LSS_{2500}}$ is typically higher than $\sigma_{LSS_{100}}$. Figure 4d shows that the differences between $\sigma_{LSS_{2500}}$ and $\sigma_{LSS_{100}}$ are smallest for the very high and low $\sigma_{LSS_{100}}$. However, Fig. 4c shows that the ensemble averages $\overline{LSS}_{2500}$ and $\overline{LSS}_{100}$ are similar, as expected from the additional zero-mean predictor variable perturbation. The values of $\overline{LSS}_{2500}$ are slightly smaller than those of $\overline{LSS}_{100}$, except for very small $\overline{LSS}$ ($< 0.1$).

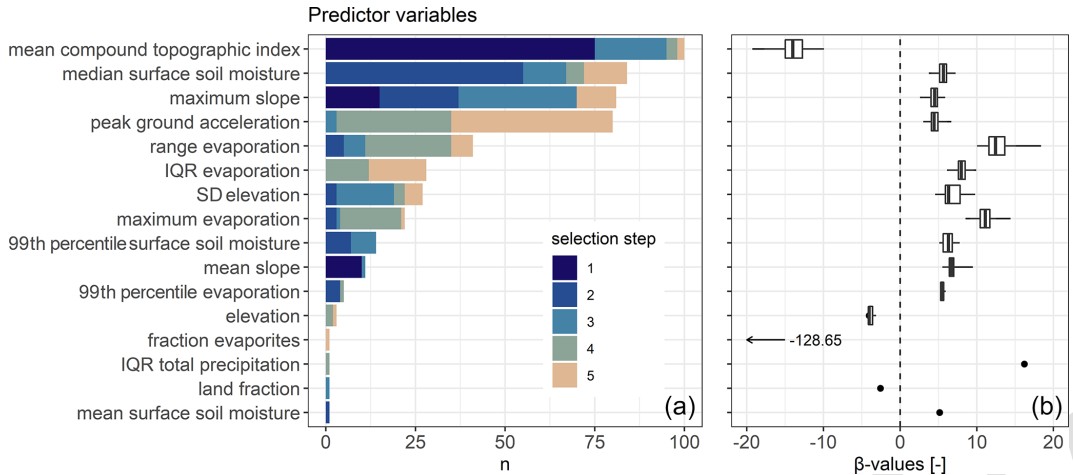

**Figure 2. (a)** Frequency of selected predictor variables and **(b)** corresponding $\beta$ values. The five best predictor variables (out of 77; see Table 1) are determined using stepwise forward selection for each MELR model equation ($n = 100$). Colors indicate at which selection step (1–5) the predictor variable was selected. Boxplots for $\beta$ values are based on the $n$ values of panel **(a)**, independent of the selection step. Whiskers extend from minimum to maximum $\beta$ values. Where $n = 1$, boxplots are replaced by a point.

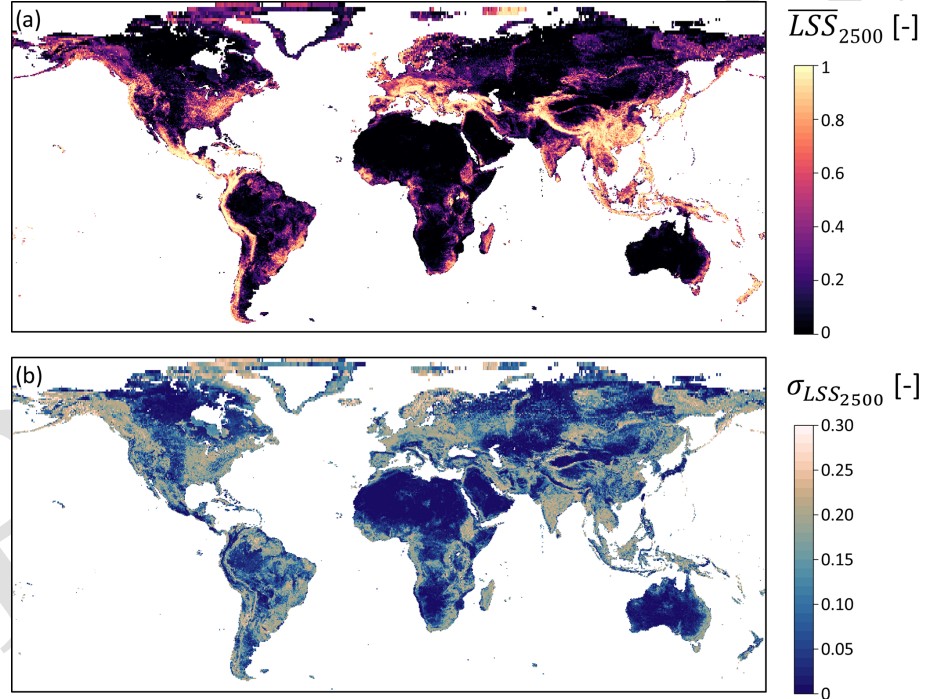

**Figure 3. (a)** Ensemble average LSS ($\overline{LSS}_{2500}$) and **(b)** standard deviation ($\sigma_{LSS_{2500}}$) at 36 km resolution. White areas denote missing values (water bodies, ice). Seemingly larger grid cells in the north are characteristic of the EASEv2 grid projection.

Figure 7 shows boxplots of the AUC values for individual members of the CV ensemble ($LSS_{100}$) and the full ensemble ($LSS_{2500}$) compared against the according CV test subsets, as well as the independent validation inventories. Note that $LSS_{100}$ is a subset of $LSS_{2500}$. The median AUC value is lower for $LSS_{2500}$ than for $LSS_{100}$ for all reference data. Despite this shift, a number of the $LSS_{2500}$ ensemble members also perform better than any of those from $LSS_{100}$. The intention is not for the individual ensemble members to have the best prediction but rather for the ensemble average $\overline{LSS}$ to be best: clearly the ensemble mean performs better than the majority of the individual ensemble members. We find AUC values for these $\overline{LSS}_{2500}$ and $\overline{LSS}_{100}$ (dots on the figure) to be practically the same (Fig. 4c)

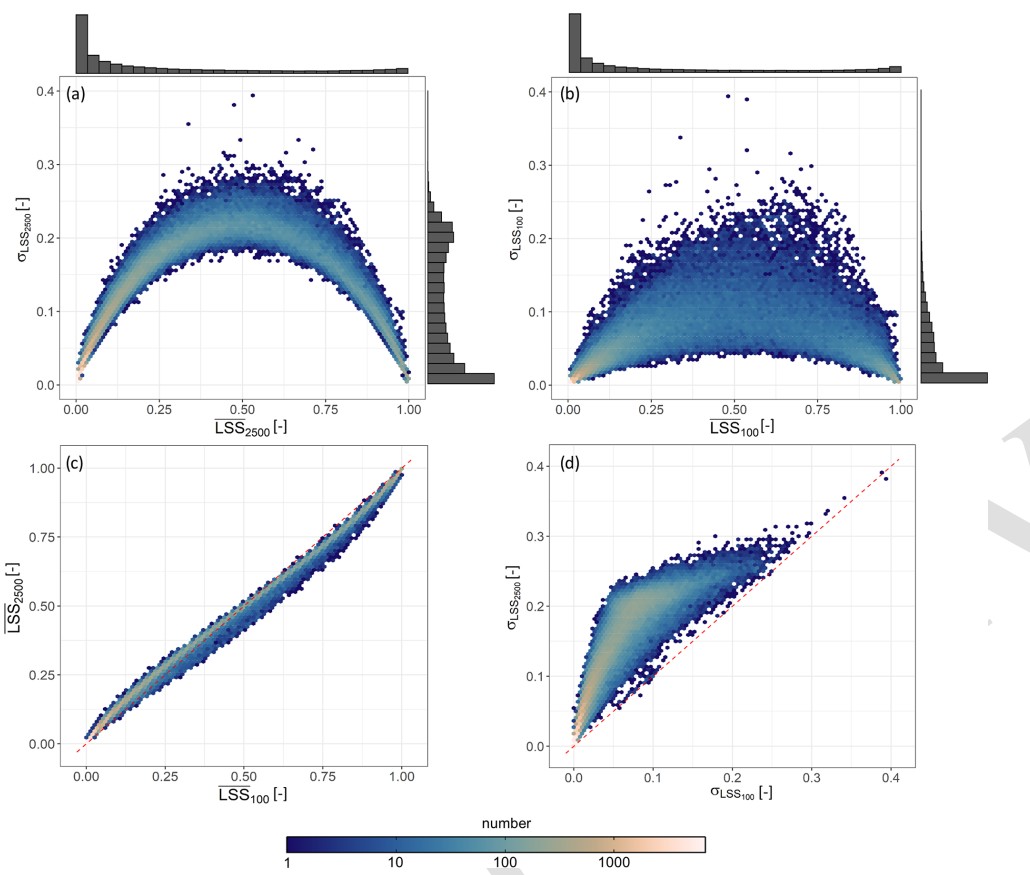

**Figure 4. (a, b)** Ensemble standard deviation LSS ($\sigma_{LSS}$) versus ensemble average ($\overline{LSS}$) of **(a)** the full ensemble ($LSS_{2500}$) and **(b)** CV ensemble ($LSS_{100}$) with the corresponding marginal distributions. The marginal distributions contain values of the complete set of 112 573 land grid cells for which LSS is estimated and are scaled by their peak for visualization. **(c, d)** Comparison of the **(c)** ensemble average and **(d)** standard deviation of $LSS_{2500}$ and $LSS_{100}$. The one-to-one line (red, dashed) is shown as reference.

## 5   Discussion

### 5.1   Selected predictor variables

For the global LSS prediction of this study, the mean CTI per grid cell is the most important predictor variable. Mean and maximum slope within a grid cell are selected less often as the first predictor variable, but one of the two is still included in nearly every MELR model. We attribute the primary importance of CTI to the fact that our model is trained with data from hydrologically triggered landslides (Kirschbaum et al., 2010, 2015), which do not uniquely occur on steep slopes. The CTI intrinsically contains information on the potential hydrological conditions of the site (through the catchment area) as well as its slope. In line with our study, Emberson et al. (2022) found that the CTI is a strong predictor of rainfall-induced landslides for a number of inventories in the tropics and subtropics. Earlier global LSS maps by Nadim et al. (2006), Hong et al. (2007) and Stanley and Kirschbaum (2017) primarily used slope information, while Lin et al. (2017) use relative relief. The latter is comparable

to the standard deviation of elevation, which is selected in more than 25 % of the models of our study.

Long-term median surface soil moisture was most frequently selected as the second predictor variable and part of more than 80 % of all models. The positive connection to LSS reflects the fact that hydrologically triggered landslides mostly occur in humid regions where the soil is often wet and rainfall can more easily destabilize a slope. The close relation between surface soil moisture and rainfall characteristics is probably the reason for its preferred selection compared to deeper layer soil moisture variables. The high correlation between surface soil moisture and both rainfall and deeper layer soil moisture variables prevents the latter two from being selected during one model creation (see Sect. 3.2). The preference for median surface soil moisture over average rainfall might be due to the less extreme values in soil moisture (quasi-normal distribution) compared to the highly non-normal distribution of rainfall but could also reflect that surface soil moisture intrinsically contains additional information on the soil characteristics. It can be interpreted as a proxy or integrator of rainfall patterns, soil, and possibly also

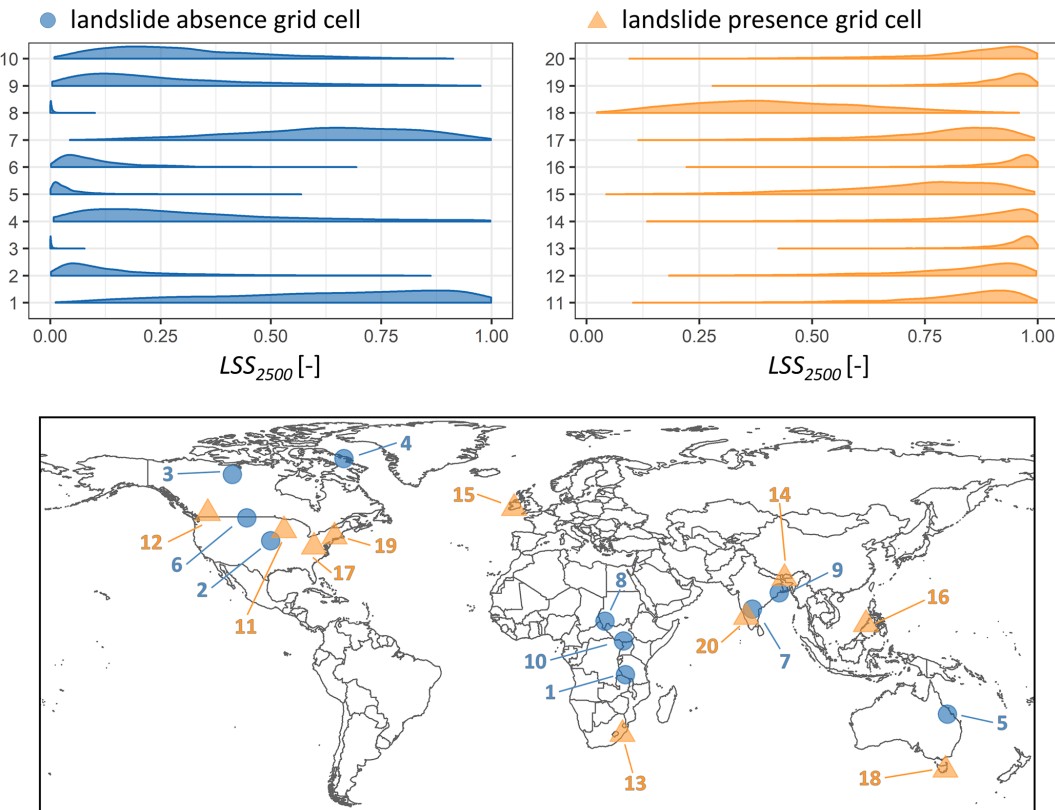

**Figure 5.** Distribution of ensemble member LSS values (LSS$_{2500}$) within sample grid cells for select landslide presence (light orange triangle on map) and absence (blue circle on map) grid cells. Please note that the distributions (top) all contain 2500 LSS ensemble members and are merely scaled by their peak to avoid overlaying (large peak) or invisible (small peak but wide distribution) curves.

vegetation characteristics. Similar to surface soil moisture, a positive relation of LSS is found for the (inter-quartile) range of evaporation. This accounts for regions with strong seasonality in rainfall and in the associated evaporation over wet soils.

In earlier global LSS maps, Nadim et al. (2006) and Lin et al. (2017) included information on the soil moisture in the form of a soil moisture index by Willmott and Feddema (1992) that distinguishes wet and dry climates. Lin et al. (2017) found this index to be the most important predictor variable. Broeckx et al. (2018) include climatological average annual rainfall as a predictor variable for LSS over Africa. At the global scale, the use of climatological statistics of hydrometeorological variables for LSS has not been tested before. It is important to note that such long-term statistics are meant to remain constant in time for global LSS estimation (by definition), but they also offer the possibility to recompute and refine LSS estimates in an era of climate change.

We did not find significant contributions of lithological predictor variables. For Africa, Broeckx et al. (2018) found a (limited) impact of the presence of unconsolidated sediments and siliclastic sedimentary rocks on LSS. Stanley et al.

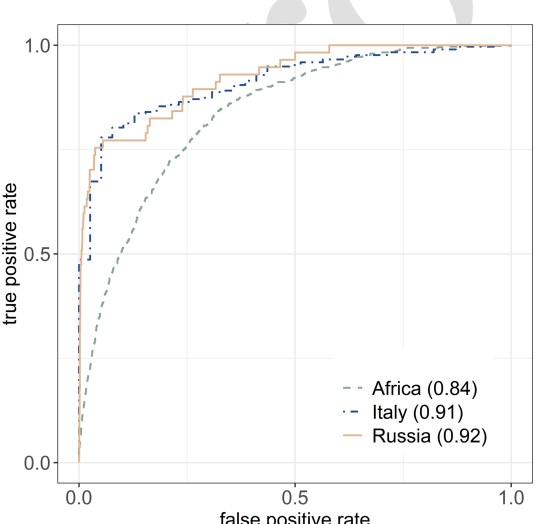

**Figure 6.** ROC curves of full ensemble average LSS ($\overline{\text{LSS}}_{2500}$) for validation inventories from Russia, Italy and Africa. Corresponding AUC values are denoted in brackets.

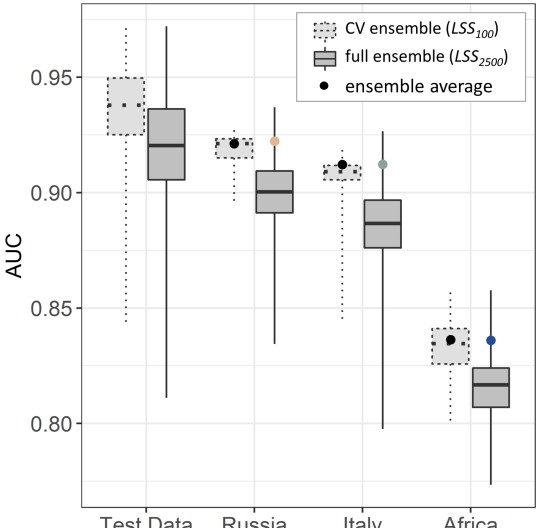

**Figure 7.** Distribution of AUC for model fitting performance (test data) and model prediction performance (based on independent validation inventories from Russia, Italy and Africa). Boxplots are shown for CV ensemble members ($LSS_{100}$) and full ensemble members ($LSS_{2500}$, including CV ensemble members), with whiskers extending from minimum to maximum AUC. AUC values for ensemble averages are displayed as points (black: $\overline{LSS}_{100}$; colored: $\overline{LSS}_{2500}$). The latter correspond to the ROC curves shown in Fig. 6.

(2021) found the lithology (regrouped from GLiM) to be the least important factor. While local lithology plays a vital role for landslide occurrence, the large data uncertainty and often very broad definitions (as for example elaborated by Campforts et al., 2020, in a different context) hinder meaningful contributions to LSS assessment, even for smaller-scale studies. This might also explain why, instead, PGA was favored as a proxy for structural weakening during the variable selection. The one-time selection of the fraction of land within a grid cell, with a negative $\beta$ value assigned, reflects that coastal or shore areas with a low land fraction are more prone to landslides (higher LSS).

Overall, the selected predictor variables and the assigned $\beta$ values are in line with general geomorphologic understanding and previous studies. We acknowledge, however, that not all possible predictors for landslides were included in the analysis. For example, land cover and land use were not explicitly included (although they are implicitly included in the climatological statistics of soil moisture, runoff and evaporation). Forest has been found to be less susceptible to landslides than non-forested areas in some regional studies (Sidle and Bogaard, 2016; Knevels et al., 2021; Depicker et al., 2021; Steger et al., 2020), although Stanley and Kirschbaum (2017) pointed out that landslides are also simply more easily observed in non-forested areas. Land cover and land use change, e.g., deforestation and urbanization (possible slope undercutting and changes in the natural drainage system of

hillslopes), are also known to increase propensity for landslides (Dille et al., 2019; Depicker et al., 2021). Stanley and Kirschbaum (2017) include forest loss and presence of roads as predictor variables for their global susceptibility map. With the expanding human presence, such predictor variables would require temporal updates and need further research for global applications.

## 5.2 Full ensemble results

The spatial patterns of the full ensemble average LSS ($\overline{LSS}_{2500}$; see Fig. 3) agree well with those of the categorical LSS maps by Stanley and Kirschbaum (2017) at 1 km resolution and Lin et al. (2017) at 0.5° resolution, shown in Fig. 8a and b. Figure 8 c and d show the distribution of the continuous 36 km $\overline{LSS}_{2500}$ per LSS class of these two reference maps. In comparing the maps, we find a larger area covered by high $\overline{LSS}_{2500}$ for example in the Eastern United States, Latin America, Mediterranean Europe, India, Southeast Asia and New Zealand. At the same time, $\overline{LSS}_{2500}$ shows much less variation than the map by Stanley and Kirschbaum (2017) within large deserts (Sahara, Arabian Peninsula and central Australia). This might be a result of the coarser spatial resolution but is also attributable to the fact that $\overline{LSS}_{2500}$ is strongly governed by hydrological predictor variables apart from the typical geomorphological ones. With a very large proportion of the lowest LSS class, Lin et al. (2017) have even less variation within these areas than $\overline{LSS}_{2500}$.

These realistic spatial distributions of $\overline{LSS}_{2500}$ are supported by the AUC values calculated for this ensemble average (dots in Fig. 7). The lower AUC value for Africa can be attributed to the fact that the inventory comprises also very old landslides from very different climatic conditions. In general though, these AUC values are in line with those of Stanley and Kirschbaum (2017), who reported AUC between 0.6 and 0.9, and Lin et al. (2017), who reported AUC around 0.9.

Figure 5 shows that the distributions of LSS ensemble members within one grid cell could have a very wide range. Even though in this figure we only selected locations within English-speaking countries and excluded unreliable absence grid cells (see Sect. 2.1), it is still possible that an absence grid cell could experience a landslide, even if none has been reported in the GLC. Prominent examples of this are absence grid cells 1 and 7, located in the East African Rift and India, respectively. Both grid cells have no reported landslide but very wide LSS distributions, with relatively high LSS values. This discrepancy between prediction and observation could indicate the need to visit this location for landslide research. At the same time, landslide presence grid cell 18 also has a very wide LSS distribution with a rather low average. This could either indicate that a non-hydrological process caused the landslide (misclassification) or that specific unrepresented features are present within the grid cell area. Overall, we find an average $\overline{LSS}_{2500}$ of 0.18 (0.82) for landslide absence (presence) grid cells (as displayed in Fig. A1),

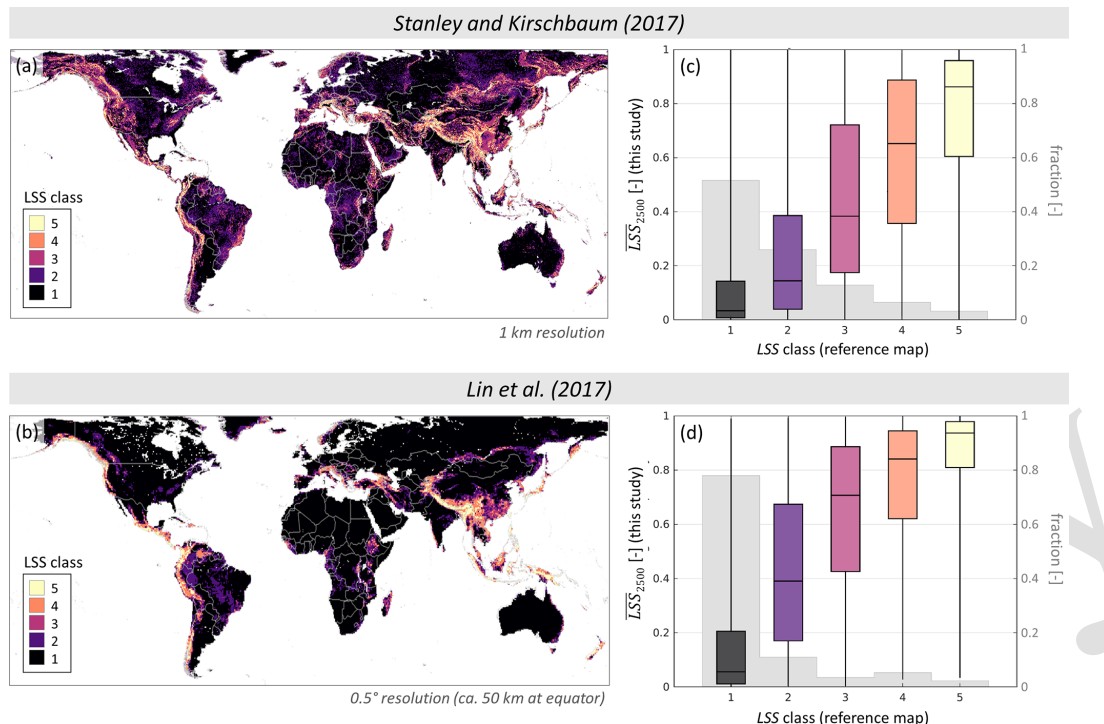

**Figure 8.** Comparison of $\overline{LSS}_{2500}$ against existing global categorical LSS maps by **(a)** Stanley and Kirschbaum (2017) and **(b)** Lin et al. (2017). Boxplots show $\overline{LSS}_{2500}$ values extracted from the nearest 36 km grid cell for each **(c)** 1 km and **(d)** 0.5° grid cell in the reference map per LSS class. Whiskers extend from minimum to maximum $\overline{LSS}_{2500}$. Boxplots are underlain with the fractions of the reference map LSS classes (grey). Note that both reference maps start off from continuous LSS values but use very different thresholds for the class definitions: Stanley and Kirschbaum (2017) set breakpoints at [0.11, 0.49, 0.67, 0.75], defined so that each category contains twice as many grid cells as the next highest, whereas Lin et al. (2017) set breakpoints at [0.4, 0.6, 0.7, 0.9], following Guzzetti et al. (2006) and Van Den Eeckhaut et al. (2012).

which makes us confident in our classification of these grid cells.

Calculating the ensemble standard deviation of these distributions ($\sigma_{LSS_{2500}}$) is a good measure of total predicted uncertainty associated with the $\overline{LSS}_{2500}$ for one grid cell. The $\sigma_{LSS_{2500}}$ is typically small for distributions at either end of the LSS interval (0, 1), resulting in the parabolic pattern as displayed in Fig. 4a–b. This pattern has also been found for local assessments (Guzzetti et al., 2006; Depicker et al., 2020) and holds for Broeckx et al. (2018) over Africa as well (visual comparison of two maps). The reasons for this relationship between $\overline{LSS}_{2500}$ and $\sigma_{LSS_{2500}}$ are twofold: (i) the classification algorithm works best for extreme environmental conditions, such as very steep slope or completely flat areas and has a strongly nonlinear, asymptotic behavior (logistic regression), and (ii) the predictions are limited to the interval (0, 1), restraining the opportunity for deviations at the extremes to one side. A comparison of $\sigma_{LSS_{2500}}$ with independent global estimates is currently not possible for lack of uncertainty estimates (Nadim et al., 2006; Hong et al., 2007; Stanley and Kirschbaum, 2017; Lin et al., 2017). However, a comparison with the standard deviations retrieved during the process of blocked random CV for the continental LSS map of Africa by Broeckx et al. (2018) (i.e., not accounting for the total uncertainty) reveals that the patterns are very similar but with less (more) variation in $\sigma_{LSS_{2500}}$ for the very arid (humid) regions.

### 5.3 Impact of input perturbations

In this study, we add predictor variable perturbations to the CV approach in order to obtain a more reliable estimate of the total predicted uncertainty from the resulting full ensemble. By design, the zero-mean input perturbation does only marginally affect the ensemble $\overline{LSS}$ (see Fig. 4). Slightly increased (decreased) $\overline{LSS}_{2500}$ at the lower (upper) limits can be attributed to the resampling of predictor variable values if they exceed the definition interval of rescaled predictor variables (0, 1). Overall, this introduced bias remains small.

The AUC analysis (Fig. 7) shows that the ensemble averages perform much better than individual ensemble members and that $\overline{LSS}_{2500}$ and $\overline{LSS}_{100}$ perform equally well. Not shown is that the BS (Eq. 2) decreases (i.e., improves) for $LSS_{2500}$ in comparison to $LSS_{100}$ where LSS is not very close to the observation already (landslide presence and absence). This effect is, however, not visible in the AUC com-

parison (spatial accuracy) for the validation data in Russia, Africa and Italy because the grid cells with BS improvement only make up for $\sim 8\%$, $\sim 9\%$ and $\sim 18\%$ respectively. The AUC values of ensemble averages remain practically the same, and an LSS model without predictor perturbations would hence suffice for a general insight in the global spatial LSS pattern.

That the individual ensemble member LSS maps of $LSS_{2500}$ (based on perturbed variables) have lower median AUC values than $LSS_{100}$ is logical: the model equations are tailored to the original predictor variable values so that they are optimally combined into an LSS prediction. Any change of these variables could deteriorate the outcome. This is, however, not a lack in quality of the ensemble but rather a side effect. We do not use the individual ensemble members but their average as an LSS prediction, for which we find practically unchanged spatial accuracy between CV ensemble and full ensemble.

By tuning the predictor variable perturbations to match the total ensemble predicted uncertainty to the observed actual uncertainty, we are able to provide statistically reliable uncertainty estimates for the predicted average LSS, even in places where landslide observations are unavailable. As stated before, this optimized spread is introduced to the input variables but does not actually reflect the input errors only: it also compensates for other uncertainty sources that are not specifically addressed, including spatial representativeness error, and uncertainties introduced by heuristic decisions along the way, such as the choice of the statistical model. Explicitly accounting for these error sources would require dedicated analyses (as for example conducted by Depicker et al., 2020). Because Zêzere et al. (2017) found that the choice of spatial mapping unit influences LSS estimates stronger than the choice of statistical model, we do not expect that our results would fundamentally change for approaches other than MELR. Future research could explore the additional information, such as landslide sizes, types or the frequency of occurrence per grid cell instead of reducing the data to landslide presence and absence. For the latter, one would need to find ways to counteract the English-language and economic bias of the GLC, which is more pronounced when using the actual number of reports instead of the presence–absence method chosen in this study.

## 6   Conclusions

This study presents the first global landslide susceptibility (LSS) map directly developed to be compatible with satellite soil moisture products retrieved from passive microwave sensors, i.e., at a spatial resolution of 36 km. The novel method of combining B-CV and predictor variable perturbations results in a reasonable assessment of the associated total predicted uncertainty. For each grid cell, we estimate 2500 individual LSS values (full ensemble) that are sum-

marized by the ensemble average LSS ($\overline{LSS}$) and standard deviation ($\sigma_{LSS}$, i.e., the uncertainty). Together, these LSS statistics can provide unprecedented information for subsequent global probabilistic spatiotemporal landslide modeling and statistical combination of the LSS and soil moisture estimates, each with their respective uncertainties. Furthermore, the LSS maps have the potential to discern areas that deserve more attention for landslide detection.

A mixed effects logistic regression (MELR) is used as the model structure to relate environmental predictor variables to spatial landslide likelihoods. The objectively selected predictor variables are mainly related to slope and hydrology, in line with the expectations for hydrologically triggered landslides. The odds of landslide occurrence were found to (i) decrease with increasing compound topographic index (CTI), which depends on the ratio of catchment area and slope, and (ii) increase with increasing slope, peak ground acceleration (PGA), and long-term climatological statistics of surface soil moisture (median and 99th percentile) or range of evaporation. The inclusion of long-term statistics of hydrometeorological variables enables future investigations into possible shifts in LSS due to climate change.

The map of the full ensemble $\overline{LSS}$ reproduces global patterns of LSS as presented in previous global studies well. The performance assessment yields area under the ROC curve (AUC) values of 0.92, 0.91 and 0.84 for independent data from Russia, Italy and Africa, respectively. The uncertainty $\sigma_{LSS}$ is largest for intermediate $\overline{LSS}$. High predicted LSS at (reliable) landslide absence grid cells might furthermore indicate regions that could benefit from future landslide detection and research.

For the ensemble perturbations of the selected predictor variables we use a perturbation magnitude of 15% to 20%, linearly proportional to the variation of elevation within a grid cell. The magnitude is chosen to match the total predicted ensemble uncertainty with the observed actual uncertainty relative to data from the Global Landslide Catalog (GLC). Adding these perturbations does not linearly propagate into the ensemble spread due to the asymptotic nature of logistic regression. It increases the ensemble spread for locations of intermediate $\overline{LSS}$ while having negligible impact where $\overline{LSS}$ is close to its lower or upper limit. The ensemble $\overline{LSS}$ and its spatial accuracy (AUC) remain practically unchanged by the ensemble perturbations, but AUC values of these average predictions are always much better than that of individual ensemble realizations. In short, these novel methods explicitly focus on the uncertainty quantification. The availability of global reliable uncertainty estimates is an unprecedented new contribution to the suite of global LSS maps, and it will support stochastic landslide hazard modeling.

# Appendix A

## A1 Landslide absence sampling

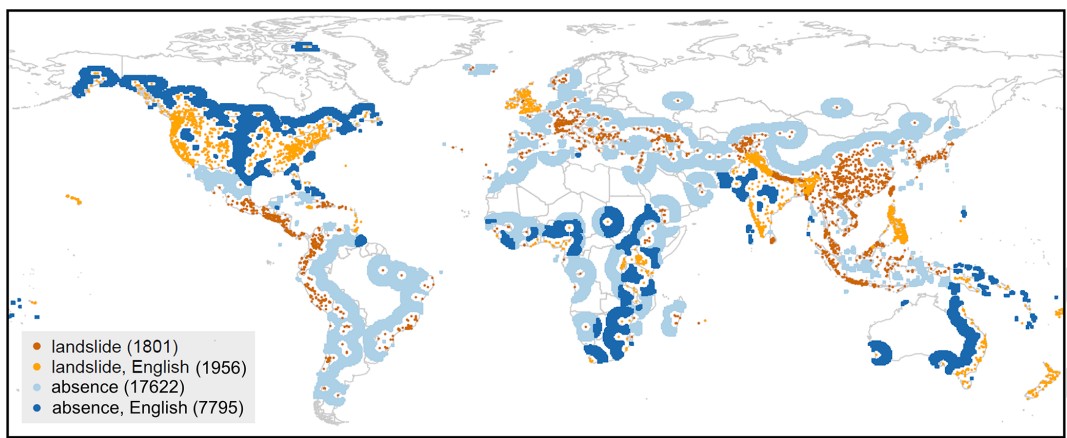

**Figure A1.** Spatial distribution of landslide presence (shade of orange) and absence (shade of blue) grid cells at 36 km resolution, for English-speaking countries (light orange and dark blue) and non-English-speaking countries (dark orange and light blue). White indicates grid cells that are excluded during the model creation process (buffer and maximum radius around landslide location; see Sect. 2.1). The numbers are the sum of each subgroup of grid cells.

Figure A1 shows the $N_{LS} = 3757$ landslide locations based on data from the GLC aggregated to the 36 km EASEv2 (Sect. 2.1). Landslide absence grid cells are sampled between a minimum (buffer) and maximum distance around known landslide locations ($N_{noLS} = 25\,417$). These distances can be based on either heuristic choices (Van Den Eeckhaut et al., 2012; Lin et al., 2017; Knevels et al., 2020) or empirical approaches (Zhu et al., 2017; Nowicki Jessee et al., 2018; Lucchese et al., 2021).

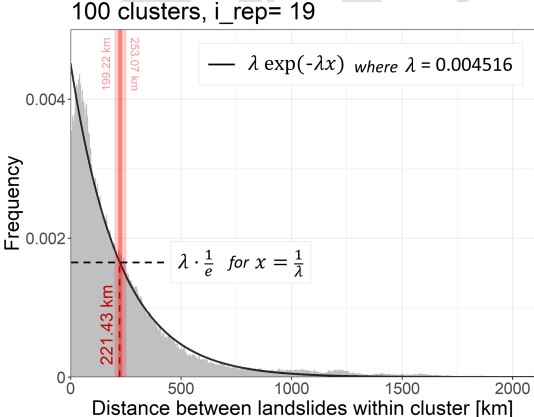

**Figure A2.** Histogram of distances [km] between landslides within a $k$-means cluster (for 100 clusters across the globe) of the GLC (grey) and Poisson exponential fit (black line) to retrieve the characteristic landslide distance (red). The red dashed line indicates median characteristic landslide distance from 50 repetitions of the $k$-means clustering, with the smallest and largest characteristic distance indicated by the light red bar and numbers at top.

For our global study, we set a buffer based on the probability for any two landslide locations from the GLC to be reported within a specific distance interval for 100 spatially defined clusters ($k$-means clustering Lloyd, 1982, on latitude and longitude). Figure A2 shows that the frequency of encountering two landslide locations decreases for larger distances and can be characterized by a Poisson exponential fit. In line with the definition of autocorrelation length (Gaspari and Cohn, 1999), we define the "characteristic distance" between two landslides as the distance where the probability to meet another landslide drops by $1/e$. We use this characteristic distance of 221.43 km or ca. six 36 km grid cells (median of characteristic distances retrieved for 50 repetitions of the clustering) as a buffer around landslide locations. The maximum distance around a landslide is subsequently defined as 2.5 times this characteristic distance (553.58 km, $\sim$ 15 grid cells), borrowing from the data assimilation community where 2.5 times the autocorrelation length is a measure for absence of correlation (Gaspari and Cohn, 1999; De Lannoy, 2006; De Lannoy et al., 2010).

Landslide absence grid cells are hence selected from 7 to 15 grid cells around a landslide presence grid cell (blue grid cells in Fig. A1). These distances are inevitably much larger than those found in literature for finer-scale studies, because autocorrelation lengths are scale-dependent and the retrieved characteristic distance is influenced by the spatial extent, or the definition of the clusters in our case.

## A2 Input perturbation and optimization

For a reliable assessment, the total ensemble predicted uncertainty of the obtained ensemble average $\overline{LSS}$ map ide-

ally should match the observed actual uncertainty. The first can be defined for a single location by the standard deviation ($\sigma$) among the LSS ensemble members ($\text{LSS}_i$, with $i = 1, \ldots, N_{\text{ens}}$), as also displayed in Sect. 4 and Fig. 3. Similarly, it is possible to assess the according variance ($\sigma^2$), referred to as ensemble spread (ensp):

$$\text{ensp} = \frac{1}{N_{\text{ens}}} \sum_{i=1}^{N_{\text{ens}}} (\text{LSS}_i - \overline{\text{LSS}})^2. \tag{A1}$$

The observed actual uncertainty at a single location is defined as the difference between $\overline{\text{LSS}}$ and the aggregated landslide observations from the GLC ($o$), referred to as the ensemble skill (ensk):

$$\text{ensk} = (\overline{\text{LSS}} - o)^2, \tag{A2}$$

where $o$ is 1 (0) in case of a landslide presence (absence) grid cell. The smaller ensk, the closer the predicted $\overline{\text{LSS}}$ to the observation. This is essentially a Brier score (see Eq. 2) for one single grid cell. (Wilks, 2011).

The optimization of the uncertainty estimates entails tuning of ensp to match ensk. In this study, this is done by varying the perturbation magnitude that is added to the input variables (see Sect. 3.2). Talagrand et al. (1997) defined spread–skill relationships that allow us to verify the statistical consistency between the assumed uncertainty (chosen perturbation) and the actual observed uncertainty based on the ergodicity principle. Over a large number of realizations, i.e., for large enough ensembles, $\langle \text{ensk} - \text{ensp} \rangle \to 0$ or

$$\frac{\langle \text{ensk} \rangle}{\langle \text{ensp} \rangle} \to 1 \Leftrightarrow \log \left( \frac{\langle \text{ensk} \rangle}{\langle \text{ensp} \rangle} \right) \to 0, \tag{A3}$$

where $\langle . \rangle$ denotes the average. In most hydrological or meteorological applications, this is the temporal average within one grid cell. As this is not applicable for the static LSS data, we consider (i) spatial averages $\langle \text{ensk} \rangle / \langle \text{ensp} \rangle$ per LSS interval as well as (ii) the distribution of individual ensk/ensp per grid cell. Both should only be performed over grid cells with reliable information about landslide presence or absence (see Appendix A1). Note that this definition of $\langle \text{ensk} \rangle$ corresponds to the definition of the BS as given in Sect. 3.2.

We tested various magnitudes of perturbations to the rescaled predictor variables either by using (i) a globally constant standard deviation or (ii) a standard deviation proportional to the topographic complexity (i.e., the variation within a grid cell, here the standard deviation of elevation). A range of possible perturbation options was tested for a partial ensemble ($\text{LSS}_{125}$, i.e., no repetition of landslide absence sampling as illustrated in Fig. 1). Figure A3 shows $\log(\langle \text{ensk}_{\text{LSS}_{125}} \rangle / \langle \text{ensp}_{\text{LSS}_{125}} \rangle)$ for 10 intervals of $\overline{\text{LSS}}_{125}$ and two examples of constant and linear perturbations. Adding any of the four perturbations brings $\log(\langle \text{ensk}_{\text{LSS}_{125}} \rangle / \langle \text{ensp}_{\text{LSS}_{125}} \rangle)$ values closer to zero, i.e., improves the spread–skill relationship, compared to results

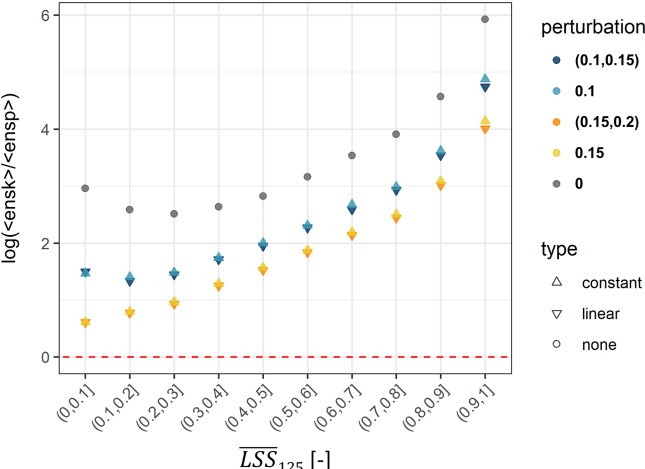

**Figure A3.** Spread–skill relationship $\log(\langle \text{ensk} \rangle / \langle \text{ensp} \rangle)$, stratified per ensemble average LSS ($\overline{\text{LSS}}_{125}$). The optimum of 0 is indicated by the red dashed line. Shapes indicate the type and colors the magnitude (constant) and interval (linear) of perturbation.

without a perturbation ($\text{LSS}_5$, single CV ensemble). Linear perturbations introduce larger spread in areas of higher $\overline{\text{LSS}}_{125}$ resulting in $\log(\langle \text{ensk}_{\text{LSS}_{125}} \rangle / \langle \text{ensp}_{\text{LSS}_{125}} \rangle)$ closer to zero than constant perturbations and are therefore preferred here.

We further analyze the distribution of individual ensp and ensk across all grid cells in Fig. A4 (top), stratified for landslide presence and absence. Ideally, ensp versus ensk should stay close to the one-to-one line. Adding a perturbation to the predictor variables (Fig. A4 c in comparison to a) nudges the distribution in this direction but fails to do so for large ensk: a large ensk results from a large difference between $\overline{\text{LSS}}_{125}$ and landslide observation ($o$), and it often coincides with very small ensp. This can be attributed in part to the incompleteness of the GLC (missing observations in a very susceptible area) and the coarse spatial resolution of this study (one very susceptible location surrounded by dominantly non-susceptible area within grid cell). Note also that the logistic regression (see Eq. 1) does not linearly propagate the perturbations of predictor variables into the resulting LSS values, especially not at the edges of the definition interval (0, 1). Accepting this tail of the distribution as an unavoidable characteristic, we further analyze the histogram of grid-cell-wise log(ensk/ensp) as displayed in Fig. A4b and d. An optimal perturbation would result in median log(ensk/ensp) close to zero and a small inter-quartile range (IQR). We therefore define the optimal perturbation for a minimum Euclidean distance ($d$) between the point (median|IQR) and (0|0), averaged over the distribution of observed landslide presences and absences ($o = 0, 1$):

$$\overline{d} = \frac{1}{2} \sum_{o=0,1} (\text{median}^2 + \text{IQR}^2)_o. \tag{A4}$$

The $\overline{d}$ for a range of possible linear perturbation options for $LSS_{125}$ is summarized in Fig. A4e. The optimal perturbation (smallest $\overline{d}$) scales the applied standard deviation according to topographic complexity, represented by the standard deviation of elevation within a grid cell, between $(0.15, 0.2)$, i.e., between 15 % and 20 %. Fine tuning of the standard deviation is left for future work but could involve other variables or transformations thereof or different amounts of perturbations per predictor variable.

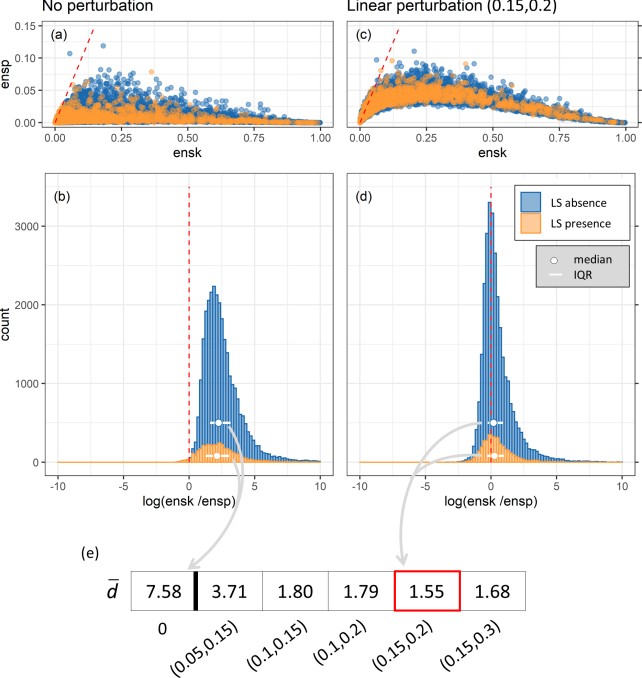

**Figure A4.** Spread–skill relationship per grid cell with the optimum indicated by the red dashed lines: **(a, c)** scatter plots of ensk against ensp; **(b, d)** histograms of log(ensk/ensp), stratified for landslide presence and absence (between buffer and maximum distance). **(e)** Summary of the average Euclidean distance $\overline{d}$ for all applied linear perturbations with the optimum framed in red. Shown are results for panels **(a)**–**(b)** without perturbation of predictor variables ($LSS_5$) and panels **(c)**–**(d)** for linear perturbation of predictor variables within the interval $(0.15, 0.2)$ ($LSS_{125}$). In other words, panels **(a)**–**(b)** account for model uncertainty alone, whereas panels **(c)**–**(d)** account for the total uncertainty (see Fig. 1).

**Abbreviations**

| | |
|---|---|
| AIC | Akaike information criterion |
| AUC | area under the ROC curve |
| B-CV | blocked random CV |
| BS | Brier score |
| CLSM | Catchment Land Surface Model |
| CTI | compound topographic index |
| CV | cross validation |
| DEM | digital elevation model |
| EASEv2 | Equal-Area Scalable Earth version 2 |
| ensk | ensemble skill |
| ensp | ensemble spread |
| GLC | Global Landslide Catalog |
| GLiM | Global Lithological Map |
| GRIP | Global Roads Inventory Project |
| IQR | inter-quartile range |
| LHASA | Landslide Hazard Assessment for Situational Awareness |
| LRC | Landslide Reporter Catalog |
| LSS | landslide susceptibility |
| MELR | mixed effects logistic regression |
| MERRA-2 | Modern-Era Retrospective analysis for Research and Applications, Version 2 |
| $N_{LS}$ | number of landslide locations, i.e., landslide presence grid cells |
| $N_{noLS}$ | number of landslide absence grid cells |
| PGA | peak ground acceleration |
| RND | average road network density |
| ROC | receiver operating characteristic |
| SMAP | Soil Moisture Active Passive |
| SMOS | Soil Moisture and Ocean Salinity |
| SRTM | Shuttle Radar Topography Mission |
| USGS | United States Geological Survey |
| VIF | variance inflation factor |

*Code and data availability.* For most of the landslide and environmental predictor data, we refer the reader to the provided sources. Source code and climatological statistics of hydrological parameters in netCDF format can be obtained by contacting the authors. The resulting full LSS ensemble is available on Zenodo (https://doi.org/10.5281/zenodo.6893230, Felsberg et al., 2022).

*Author contributions.* AF designed the LSS assessment setup, created the code and conducted the analysis, supervised by GJMDL and JP along the way. GJMDL provided scientific guidance for all steps of this study, with special focus on the input perturbation and optimization. MB provided guidance for the CV approaches. JP and MV provided topical expertise for interpretation of results. All co-authors provided guidance on the study's content and contributed to the paper.

*Competing interests.* The contact author has declared that none of the authors has any competing interests.

*Acknowledgements.* We thank Thomas Stanley for providing data, feedback and recommendations. We also thank Jente Broeckx for his input, Bianca Drepper for the support in the data preparations and two anonymous referees for their valuable review comments. We additionally thank Luca Brocca for being part of the advisory committee of AF. The computational resources (high-performance computing) and services used in this work were provided by the VSC (Flemish Supercomputer Center).

*Financial support.* Anne Felsberg was funded by the Fonds Wetenschappelijk Onderzoek (grant no. FWO-G0C8918N). VSC usage was funded by KU Leuven (C14/16/045), FWO (1512817N) and the Flemish Government.

*Review statement.* This paper was edited by Paolo Tarolli and reviewed by two anonymous referees.

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
