# Peer review of "Estimating global landslide susceptibility and its uncertainty through ensemble modelling"

_Natural Hazards and Earth System Sciences, 2021_

## Referee Comment (RC1)

I must first apologize to the authors and to the editor for this late review. As an author, I have myself experienced how irritating it can be to wait for an overdue review. I can only mention heavy load of work as an excuse for my delayed response.

Not being myself an expert on landslide causes and occurrences, I am not in a position to comment on the parts of the paper that specifically deal with those aspects, nor actually to evaluate the paper in comparison with what has already been done on estimation of landslide susceptibility and uncertainty. But I have comments on the methodological aspects of the paper, which may be useful for all readers.

I must say I have had difficulties to clearly understand what the authors have done, as concerns both their methodological approach and the validation of the results they have obtained. I will limit myself to what are the most important points I want to stress.

1. My first question has actually to do with landslides. The authors focus their study to *hydrologically triggered landslides* (l. 68). This means that they ignore, for instance, landslides triggered by earthquakes. What is the reason for that restriction ? How can the distinction be made between different kinds of landslides, once they have occurred ? And does the Global Landslide Catalog report only hydrologically triggered landslides ? These questions may look naïve to specialists, but some appropriate information (and references) may be useful to outsiders.

2. The first purpose of the paper is to derive *LSS model equations* (ll. 66-67). The authors do not actually show any equation that is explicitly identified as such. The model equations must be equations of form (1), where the quantity P(Y = 1) is what is called LSS elsewhere. Equations of form (1), which are defined as a form of logistic regression, are used on appropriate training sets for determining, through MELR and Cross Validation, values of the parameters  $\alpha$  and  $\beta_i$  (*i*=1,...,*n*). This raises a number of questions.

*a*. What is the rationale for the logistic form of equation (1) ? What are the advantages of that specific approach ? It seems to me to be a rather arbitrary choice. In their conclusion, the authors mention *the choice of the statistical model* (l. 365) as one possible source of uncertainty in the whole estimation process. Do they refer there to Eq. (1) ? The authors give references concerning logistic regression, but some basic explanation would be useful.

*b*. In MELR, what is the criterion for quality ? Given a tentative set of values ( $\alpha$ ,  $\beta_i$ , *i*=1,...,*n*), by which measure is the corresponding fit to P(*Y* = 1) evaluated ? A simple quadratic fit, or what ?

*c*. I understand that the values P(Y = 1) in the training sets are taken in the data set built in subsection 2.1 from the GLC catalog, so that these values are restricted to 0 and 1 (absence or presence of landslides). It would *a priori* seem more appropriate to consider a quantity such as the frequency of occurrence of landslides (that would not be impossible from GLC since the latter mentions more than one landslide for a number of individual grid cells). That may not be practically possible, but it would in my mind be appropriate to mention, and preferably briefly discuss, that alternative approach.

3. *a*. Concerning the quality of the uncertainty of their LSS estimates, the authors use as diagnostic the *Receiver Operation Characteristic (ROC)* and the associated area under the ROC curve (AUC). They mention AUC values for individual members of ensembles, i.e. LSS maps (ll. 214-217, l. 275 and Fig. 7), as well as for global ensembles (ensembles of maps). The latter are all right for me, but I do not understand what AUC values for individual maps be. ROC can curves (https://en.wikipedia.org/wiki/Receiver operating characteristic) are parameterized by a threshold T, each point on the curve corresponding to a value of T. The corresponding coordinates are relative to the circumstances when the value of a given parameter is larger than T (in the context of the present paper, that parameter must be LSS). Unless all grid cells are lumped together, which does not seem to be reasonable, it does not make sense to consider the situation LSS > T on a single map. That makes sense only on an ensemble of maps, with grid cells being considered independently of each other. I may of course be mistaken as to what the authors have exactly done, but clarification is necessary.

*b*. And the reference to *one fully deterministic reference MELR equation (based on neither CV nor input perturbations)* (ll. 220-221) is confusing. Does it mean you have performed the validation on other outputs than the ones obtained from CV ? I have a similar question about the *one deterministic MELR equation* mentioned on ll. 355-356.

4. The authors write (ll. 357-359) The finding of Kalnay et al. (2006) (show) that the introduction of ensembles increases the accuracy of the prediction does not hold for our LSS modelling. This is probably due to the non-linear characteristics of logistic regression and LSS being static. I understand the authors mean that the accuracy of the mean of the output ensemble is higher than the accuracy of an individual deterministic estimate (at least statistically). From what I understand, non-linearity cannot be the problem here. Consider a process F(x) where there is uncertainty of the input x. Let  $\{x_i\}$  a sample of independent realizations of the probability distribution for x. The ensemble  $\{F(x_i)\}$  is a sample of independent realizations of the probability distribution for F(x). As such, the mean of that ensemble is the best estimate of F(x), at least in a least-square sense. That is true whether the process F is linear or not.

5. The authors write in their conclusion (ll. 373-374) ... predictor variable perturbations results in a reliable assessment of the associated total prediction uncertainty. It is of course more difficult to assess the uncertainty on an estimate than to obtain the estimate itself. But the authors' statement seems to be a bit of an exaggeration. The AUC values given in the paper do show some reliability in the assessment of the uncertainty, but no more. Actually, the amplitude of the predictor variable perturbations has been evaluated on the same data set as the LSS values. The whole process is therefore subject to some form of inbreeding, the impact of which is difficult to assess. And the authors write themselves A comparison of  $\sigma_{LSS2500}$  with independent global estimates is currently not possible for lack of uncertainty estimates (ll. 340-341). I suggest the authors soften down their concluding statement.

I would have also comments on editing aspects of the paper, but I think they are of lesser importance at this stage.

---

## Referee Comment (RC2)

I have revised the manuscript "Estimating global landslide susceptibility and its uncertainty through ensemble modelling" by Anne Felsberg and co-authors. The procedure is very interesting and it would be interesting to apply it to a smaller area where you can control better the training and the validation landslide and thematic data. The manuscript is well written and well organized but there are few major thinks nor really convincing.

1. It is possible to select the hydrologically triggered landslides from the Global Landslide Catalogue?
2. Is not clear how you have used the road network density
3. In the analysis you have mentioned model uncertainty and input uncertainty. I think the uncertainty associated to a 36-km spatial resolution grid is so large that the entire analysis is not relevant. As you mention at line 51 "*coarser input data might be less representative for local events, such as landslides*". If data are not representative the entire modelling is not representative.
4. Line 20 *LSS maps derived from environmental conditions are a fundamental tool for informing local population, city planners and decision makers both on the immanent landslide likelihood, but also about secondary effects such as major sediment sources* → this is true but very difficult to be applicable at the resolution of your analysis.
5. (Line 60) The total uncertainty is estimated by comparing the predicted average LSS against the observed presence and absence of landslides → in your case the presence/absence of landslide is related to a too coarse grid resolution.
6. Due to their generalizing nature, LSS models are however prone to uncertainty. → true but the uncertainty it is also highly related to the thematic data/landslide distribution/model used for the assessment. In your case the uncertainty associated to data and landslide distribution is more relevant than the entire modelling.
7. (line 89) When you aggregate landslide data in a landslide location, do you check that your aggregation is reliable?
8. (Line 90) Multiple landslides within the same 36-km grid cell are aggregated into one 'landslide location --> The environmental condition selected in a 36 km grid cell can be completely inappropriate and not relevant to explain the landslide.
9. (Line 111) *Absence grid cells are hence selected from grid cells 7 to 15 around a landslide occurrence* → How you can be sure that the selected conditions are not prone to landslides?
10. To compute the compound topographic index, you need the specific catchment area and the slope. How do you measure then in a 36 km grid?
11. Why do you consider the peak ground acceleration if you want to evaluate hydrologically triggered landslides?
12. (Line 161) → The mixed effects approach allows us to include a so-called 'random effect', here the random intercept α, for which we use the average road network density stratified into 6 groups (divided by the global quintile thresholds) → not clear
13. (Line 185) We group the grid cells into a total of 100 blocks according to climatological conditions within 10 predefined regions (roughly two per continent), independent of landslide absence or presence → a) this means that each block has about 75 pixel? b) What is the rational to select roughly two climatological conditions per continent? If this is not a consistent selection is not representative.
14. (Line 183) One subset consists of 20 randomly sampled 'blocks', i.e. small groups, of the 7514 grid cells selected for model creation. We group the grid cells into a total of 100 blocks according to climatological conditions within 10 predefined regions (roughly two

per continent), independent of landslide absence or presence. Within these regions, we mimic typical climatological zonations (for example that of Köppen) through k-means clustering (Lloyd, 1982) of 30-year average soil surface temperature and rainfall (see Table 1), dividing each region into 10 blocks. → Not clear the relation between the blocks and the 5 subset.

15. (Line 224) Aggregated data vs observations → Do you really think is reliable to aggregate original observations? In Italy for example you have aggregated 5438 observations in 309 points. I think the two data are completely different and infact you get very low ROC curve.

16. (Line 236) *Values of the intercept, which is part of all models, vary with road network density as part of the MELR and mostly have and average close to zero (not shown)* → can you explain better this statement?

17. Fig. 3 → how much all this complex analysis improves/enhances at worldwide scale a simple regression model applied to obtain an LLS?

18. (Line 265) *The LSS2500 map hence performs very well over Russia and Africa, while showing some difficulties to capture the patterns for Italy* → This is the situation where you have modelled 309 points aggregated from 5438 observation. Are you sure that the aggregated points are representative of the failure distribution around the world?

19. The procedure is quite complex and the real meaning between the $LSS_{2500}$ and $LSS_{100}$ is not very easy to understand

---

## Author Comment (AC1)

The authors thank the reviewers for their constructive comments. The comments are shown in regular fonts, ***our responses are in bold italic, blue fonts***. *Changes made in the manuscript are printed in italic, underlined, blue fonts.* **Our line references refer to the updated manuscript with track-changes.**
* * *
Reviewer #1
* * *
I must first apologize to the authors and to the editor for this late review. As an author, I have myself experienced how irritating it can be to wait for an overdue review. I can only mention heavy load of work as an excuse for my delayed response.

Not being myself an expert on landslide causes and occurrences, I am not in a position to comment on the parts of the paper that specifically deal with those aspects, nor actually to evaluate the paper in comparison with what has already been done on estimation of landslide susceptibility and uncertainty. But I have comments on the methodological aspects of the paper, which may be useful for all readers.

I must say I have had difficulties to clearly understand what the authors have done, as concerns both their methodological approach and the validation of the results they have obtained. I will limit myself to what are the most important points I want to stress.

***We thank the reviewer for the feedback on the methods. We improved the readability of the paper for people in various research fields, because we hope to bridge multiple disciplines in this paper and appreciate feedback from outside the landslide community.***

1. My first question has actually to do with landslides. The authors focus their study to *hydrologically triggered landslides* (l. 68). This means that they ignore, for instance, landslides triggered by earthquakes. What is the reason for that restriction ? How can the distinction be made between different kinds of landslides, once they have occurred ? And does the Global Landslide Catalog report only hydrologically triggered landslides ? These questions may look naïve to specialists, but some appropriate information (and references) may be useful to outsiders.

***Thank you for this valid question. As mentioned in lines 39-43, the susceptibility assessment carried out in this study is intended to be used with satellite soil moisture observations. Since the likelihood of earthquake triggered landslides is primarily dependent on the presence of an earthquake and its magnitude, the soil saturation is not a reliable indicator for occurrence of these landslides. For hydrologically triggered landslides, on the other hand, increased soil water content is the actual underlying condition that leads to slope failure by i) decreasing the shear strength and ii) increasing the shear stress.***

***Landslide inventories (such as the GLC) based on media reports usually take the information on the trigger from the report itself. The GLC was designed mainly for the purpose of collecting information on hydrologically triggered landslides, namely triggered by "continuous rain", "downpour", "monsoon", "flooding", "rain" and "tropical cyclone" (GLC classifiers), but also contains a small number of landslides from other triggers (less than 5%) and for about 16% triggers are unknown. We will update section 2.1 as follows:*** *"The GLC is a landslide inventory that contains information about location, date and trigger. It is originally based on media reports (Kirschbaum et al., 2010, 2015) but has recently been supplemented with the citizen science-based Landslide Reporter Catalog (LRC) data (Juang*

*et al., 2019), see Stanley et al. (2021) for details. [...] For this study, we use 12515 hydrologically triggered landslides (GLC classifiers "continuous rain", "downpour", "monsoon", "flooding", "rain" and "tropical cyclone") reported mainly between January 2007 and November 2020." (Line 95-104)*

2. The first purpose of the paper is to derive *LSS model equations* (ll. 66-67). The authors do not actually show any equation that is explicitly identified as such. The model equations must be equations of form (1), where the quantity P(Y = 1) is what is called LSS elsewhere. Equations of form (1), which are defined as a form of logistic regression, are used on appropriate training sets for determining, through MELR and Cross Validation, values of the parameters α and βi (i=1,…,n). This raises a number of questions.

a. What is the rationale for the logistic form of equation (1) ? What are the advantages of that specific approach ? It seems to me to be a rather arbitrary choice. In their conclusion, the authors mention the choice of the statistical model (l. 365) as one possible source of uncertainty in the whole estimation process. Do they refer there to Eq. (1) ? The authors give references concerning logistic regression, but some basic explanation would be useful.

*Thank you for your question. Indeed, the choice of logistic regression as statistical model was in a way arbitrary, but based on literature review and expert opinions. Many smaller scale LSS studies compare multiple statistical models and subsequently use the best performing one (for example Steger et al. 2015, Zêzere et al. 2017 or Depicker et al. 2020). Larger scale studies usually choose a priori one statistical model to limit computational time (for example Stanley and Kirschbaum 2017, Lin et al. 2017 or Broeckx et al. 2018). The findings of Zêzere et al. 2017 also show that the choice of statistical model has less impact on the final LSS results than the choice of mapping unit. We opted for a logistic regression because it is a very simple approach and one of the earliest statistical, data-driven models used for LSS (Reichenbach et al. 2018). The advantage lies in its rather robust nature, that is not as prone to overfitting as certain machine learning algorithms. As such, logistic regression is also the most commonly used technique (Reichenbach et al. 2018). We will extend the manuscript as follows:*

*"Logistic regression is the most commonly used approach for statistical LSS mapping (Reichenbach et al., 2018). It is associated with strong generalizing capabilities (Brenning, 2005), which is a necessity when working at the global scale, and it has proven to be reliable in continental to global LSS assessments (Broeckx et al., 2018; Lin et al., 2017)." (Line 164-167)*

*"Because Zêzere et al. (2017) found that the choice of spatial mapping unit influences LSS estimates stronger than the choice of statistical model, we do not expect that our results would fundamentally change for approaches other than MELR." (Line 450-452)*

b. In MELR, what is the criterion for quality ? Given a tentative set of values (α, βi,i=1,…,n), by which measure is the corresponding fit to P(Y = 1) evaluated ? A simple quadratic fit, or what ?

*For this study we used the lme4 package in R, specifically the glmer function, in which the fit is optimized based on maximum likelihood estimation, which involves a minimization of squared deviances with penalty terms. Subsequently, we evaluate the model fitting performance of each fitted equation by means of AUC on the test data, but this information is not used for optimization. We will extend section 3.1 by the following statement in hope for clarification: "We use the glmer function from the lme4 package (Bates et al. 2015) to create MELR models in R version 4.0.3 (R Core Team, 2020) where the best fitting parameters are obtained by maximum likelihood estimation." (Line 201-203).*

*In addition, note that: "a measure that is proportional to the sum of squared errors [...]" (L.177-178) is used to identify the most useful predictor variables.*

c. I understand that the values P(Y = 1) in the training sets are taken in the data set built in subsection 2.1 from the GLC catalog, so that these values are restricted to 0 and 1 (absence or presence of

landslides). It would a priori seem more appropriate to consider a quantity such as the frequency of occurrence of landslides (that would not be impossible from GLC since the latter mentions more than one landslide for a number of individual grid cells). That may not be practically possible, but it would in my mind be appropriate to mention, and preferably briefly discuss, that alternative approach.

*That is a valid point that also crossed our mind. A posterior analysis of average LSS ($\overline{LSS}_{2500}$) against the number of landslide reports per grid cell showed indeed mostly an increased median $\overline{LSS}_{2500}$ for higher reporting frequency (see Figure 1 below).*

[Figure]

*Figure 1 Ensemble average LSS ($\overline{LSS}_{2500}$) for grid cells with different number of reported landslides in the GLC, stratified for the official language denoted by the United Nations Group of Experts on Geographical Names.*

*However, this connection significantly differed between grid cells located in English-speaking countries and those of other official languages. One reported landslide in an English-speaking country is associated with lower LSS than in a non-English-speaking country. Hence, keeping the possible response values restricted to 0 or 1 also contributes to mitigating observation biases from the GLC. This posterior analysis nevertheless gives confidence that locations with a higher frequency of landslide reports have also been assigned a higher LSS by the statistical method of MELR. We will add the following short discussions of this:*

*"While we acknowledge that grid cells with more frequent landslide reporting can in general be expected to have a higher LSS, we found that the information about the frequency of landslide occurrence within a grid cell strongly mirrors biases in the landslide inventory, e.g. more landslides are reported in English-speaking countries. The aggregation, on the contrary, reduces the landslide presence reporting bias of the GLC." (Line 108-111)*

*"Future research could explore the additional information, such as landslide sizes, types or the frequency of occurrence per grid cell instead of reducing the data to landslide presence and absence. For the latter, one would need to find ways to counteract the English-language and economic bias of the GLC which is more pronounced when using the actual number of reports instead of the presence-absence method chosen in this study." (Line 452-456)*

3. a. Concerning the quality of the uncertainty of their LSS estimates, the authors use as diagnostic the Receiver Operation Characteristic (ROC) and the associated area under the ROC curve (AUC). They mention AUC values for individual members of ensembles, i.e. LSS maps (ll. 214-217, l. 275 and Fig. 7), as

well as for global ensembles (ensembles of maps). The latter are all right for me, but I do not understand what AUC values for individual maps can be. ROC curves (https://en.wikipedia.org/wiki/Receiver operating characteristic) are parameterized by a threshold T, each point on the curve corresponding to a value of T. The corresponding coordinates are relative to the circumstances when the value of a given parameter is larger than T (in the context of the present paper, that parameter must be LSS). Unless all grid cells are lumped together, which does not seem to be reasonable, it does not make sense to consider the situation LSS > T on a single map. That makes sense only on an ensemble of maps, with grid cells being considered independently of each other. I may of course be mistaken as to what the authors have exactly done, but clarification is necessary.

*Thank you for this valid question. All ROC curves and connected AUC values mentioned in this study were derived for individual LSS maps, i.e. "all grid cells lumped together", or a subset of these grid cells. In case of the ensemble, this means one AUC value is computed for each member map (or parts of it, such as Africa, Russia and Italy). The ROC allows to validate a continuous probability value against a discrete outcome (landslide presence and absence) by means of applying, as you correctly state, different threshold values T. For grid cells below (above) T, the prediction is assumed to be landslide absence (presence), and comparison against the validation data in form of a confusion matrix allows to compute the according true and false positive rate. The pairs of these rates are collected for different thresholds T and displayed in the ROC curve. "Lumping" all grid boxes of one map together, allows to evaluate the spatial accuracy and is in this way common practice for landslide susceptibility: Reichenbach et al. (2018) report that it was used as an accuracy measure in more than 20% of the 565 susceptibility studies they reviewed. We will alter section 3.3 as follows:*

*"To quantify how well a predicted LSS map represents observed landslide presences and absences, a BS can be used (see Equation 2). Alternatively, the Receiver Operating Characteristic (ROC) is commonly used as evaluation tool for categorical response values such as landslide presence and absence (Reichenbach et al., 2018). For the ROC, the true positive rate of one LSS map is displayed against its false positive rate for different possible thresholds in the continuous probability (here: LSS) that is predicted. The area under the ROC curve (AUC) is 1 for a perfect representation of the spatial LSS distribution, whereas an AUC value of 0.5 indicates that the model does not perform better than a uniform distribution.*

*Depending on the reference landslide data, the ROC analysis can be conducted for specific grid cells from a CV subset (independent data not used in the training), or from other independent landslide inventories. Here, we use landslide presence and absence information from the grid cells of the fifth CV subset (test subset, see Figure 1) to assess the model fitting performance for each LSS ensemble member map "on the go". To evaluate the final prediction performance of the complete ensemble averages and the corresponding ensemble members, we use 3 independent landslide inventories." (Line 252-272)*

b. And the reference to one fully deterministic reference MELR equation (based on neither CV nor input perturbations) (ll. 220-221) is confusing. Does it mean you have performed the validation on other outputs than the ones obtained from CV ? I have a similar question about the one deterministic MELR equation mentioned on ll. 355-356.

*We are sorry to have caused confusion with this. We wanted to evaluate whether the general introduction of an ensemble in contrast to deterministic predictions improves the accuracy of the LSS map, as found for hydrological and meteorological modelling (Kalnay et al. 2006). For this reason we created – in addition to the full ($LSS_{2500}$) and CV ensemble ($LSS_{100}$) – this one fully deterministic MELR equation. The AUC values retrieved for the resulting deterministic LSS map for Russia, Italy and Africa were then compared against the AUC values of the ensemble averages ($\overline{LSS}_{2500}$ and $\overline{LSS}_{100}$ ).*

*However, we realize that this is confusing for the reader and to some extent an unfair comparison because this fully deterministic MELR (without CV and predictor perturbations) is also trained on the complete set of landslide presence and absence grid cells (in contrast to the model training on 4 out of 5 subsets as part of CV). Since this additional comparison does not add much to the results of our study, we have decided to remove any mention of it from the manuscript:*

*We will alter section 3.3 as follows and hope that the procedure becomes more clear:* "The AUC and BS metrics can be computed for individual ensemble members (of the CV ensemble $LSS_{100}$, or the full ensemble $LSS_{2500}$, yielding a distribution of metrics) or for ensemble averages ($\overline{LSS}_{100}$ and $\overline{LSS}_{2500}$). It will be assessed whether i) an ensemble average outperforms an individual member LSS realization and whether ii) the full ensemble average with ensemble input perturbations ($\overline{LSS}_{2500}$) outperforms the CV ensemble average which does not include input perturbations ($\overline{LSS}_{100}$). This would be in line with the expectations for hydrological or meteorological models (Kalnay et al., 2006)." (Line 282-286)

4. The authors write (ll. 357-359) *The finding of Kalnay et al. (2006) (show) that the introduction of ensembles increases the accuracy of the prediction does not hold for our LSS modelling. This is probably due to the non-linear characteristics of logistic regression and LSS being static.* I understand the authors mean that the accuracy of the mean of the output ensemble is higher than the accuracy of an individual deterministic estimate (at least statistically). From what I understand, non-linearity cannot be the problem here. Consider a process F(x) where there is uncertainty of the input x. Let {$x_i$} a sample of independent realizations of the probability distribution for x. The ensemble {$F(x_i)$} is a sample of independent realizations of the probability distribution for F(x). As such, the mean of that ensemble is the best estimate of F(x), at least in a least-square sense. That is true whether the process F is linear or not.

*Thank you for this remark. Indeed you are right in that for one grid cell the mean of an ensemble {$F(x_i)$} should remain the best estimate for the process F(x), independent of the linearity of the process, and the quoted finding was an error from our side, that will be corrected.*

*Our naïve inference was that the ensemble average should also improve the spatial performance (assessed by AUC, as explained for comment 3) over that of an unperturbed realization. This, however, we did not find to be the case. A more correct assessment of the influence of the ensemble on the performance would be based on the accuracy per grid cell. A measure for this is the ensemble skill (ensk, see Equation A2), which is essentially the squared difference between (ensemble average) LSS and the observation. When averaged over a number of grid cells, ensk turns into the Brier Score (BS, see Equation 2). We do find that the BS based on intervals of CV ensemble ensk ($ensk_{LSS100}$) is improved through predictor variable perturbation (full ensemble) where $ensk_{LSS100}$ is not already close to its optimum value of 0:*

[Figure]

*Figure 2 (Top) Difference in Brier Score (BS) between the full ensemble (LSS$_{2500}$) and the CV ensemble (LSS$_{100}$), for intervals of ensemble skill of the latter (ensk$_{LSS100}$). (Bottom) Number of grid cells within the ensk$_{LSS100}$ interval.*

**These grid cells with very small ensk$_{LSS100}$, i.e. where the predictor perturbations do not improve the performance, comprise by far the largest proportion of all land grid cells. They mostly have very low LSS values close to 0, where the predictor variable perturbation was found to introduce a small bias towards higher LSS values:** *"Slightly increased (decreased) LSS$_{2500}$ at the lower (upper) limits can be attributed to the resampling of predictor variable values if they exceed the definition interval of rescaled predictor variables (0,1)." (Line 421-423)* **For landslide absence grid cells, this slightly decreases the accuracy and results in a slightly increased BS value. In the spatial accuracy assessment, it is these grid cells that strongly dominate the resulting AUC values due to their large number.**

**We will alter section 5.3 as follows:** *"The AUC analysis (Fig. 7) shows that the ensemble averages perform much better than individual ensemble members, and that $\overline{LSS}_{2500}$ and $\overline{LSS}_{100}$ perform equally well. Not shown is that the BS (Equation 2) decreases (i.e. improves) for LSS$_{2500}$ in comparison to LSS$_{100}$ where LSS is not very close to the observation already (landslide presence and absence). This effect is, however, not visible in the AUC comparison (spatial accuracy) for the validation data in Russia, Africa and Italy because the grid cells with BS improvement only make up for ~8%, ~9% and ~18% respectively. The AUC values of ensemble averages remain practically the same, and an LSS model without predictor perturbations would hence suffice for a general insight in the global spatial LSS pattern." (Line 425-431)*

5. The authors write in their conclusion (ll. 373-374) … *predictor variable perturbations results in a reliable assessment of the associated total prediction uncertainty*. It is of course more difficult to assess the uncertainty on an estimate than to obtain the estimate itself. But the authors' statement seems to be a bit of an exaggeration. The AUC values given in the paper do show some reliability in the assessment of the uncertainty, but no more. Actually, the amplitude of the predictor variable perturbations has been evaluated on the same data set as the LSS values. The whole process is therefore subject to some form of inbreeding, the impact of which is difficult to assess. And the authors write themselves *A comparison of $\sigma_{LSS2500}$ with independent global estimates is currently not possible for lack of uncertainty estimates* (ll. 340-341). I suggest the authors soften down their concluding statement.

*Thank you for this valid remark and we will soften the concluding statement. The "inbreeding" is, however, only partially true, because we also evaluate AUC values for the independent validation inventories (Russia, Africa, Italy). These were not part of the GLC, which we used for the tuning of the perturbations. We find the same tendencies in AUC spread for these validation inventories and the test data (not used for training) from the GLC (see Figure 7). We do agree though, that our assumption of reliability is strongly rooted in and connected to our trust in the tuning of the perturbations. We agree to change the text as follows: "The novel method of combining blocked random CV (B-CV) and predictor variable perturbations results in a reasonable assessment of the associated total prediction uncertainty." (Line 459-461)*

I would have also comments on editing aspects of the paper, but I think they are of lesser importance at this stage.

---

## Author Comment (AC2)

The authors thank the reviewers for their constructive comments. The comments are shown in regular fonts, ***our responses are in bold italic, blue fonts***. *Changes made in the manuscript are printed in italic, underlined, blue fonts.* ***Our line references refer to the updated manuscript with track-changes.***

Reviewer #2

I have revised the manuscript "Estimating global landslide susceptibility and its uncertainty through ensemble modelling" by Anne Felsberg and co-authors. The procedure is very interesting and it would be interesting to apply it to a smaller area where you can control better the training and the validation landslide and thematic data. The manuscript is well written and well organized but there are few major thinks nor really convincing.

***We thank the reviewer for the detailed feedback. We extended and improved explanations for many points that were mentioned as unclear and hope that this is able to increase readability and understanding.***

1. It is possible to select the hydrologically triggered landslides from the Global Landslide Catalogue?

***Please see response to comment 1 of reviewer #1.***

2. Is not clear how you have used the road network density

***Thank you for pointing this out. We will extend the mentioning of the road network density in section 2.1 to the following:*** *"To address the remaining landslide presence bias originating from more landslide reporting in frequently accessed areas, we use stratified data on the [average road network density] (including highways and all types of roads, ranging from primary to local roads) provided by the Global Roads Inventory Project (GRIP) (Meijer et al., 2018) as a random effect, explained in sect. 3.1." (Line 112-115)*

3. In the analysis you have mentioned model uncertainty and input uncertainty. I think the uncertainty associated to a 36-km spatial resolution grid is so large that the entire analysis is not relevant. As you mention at line 51 "*coarser input data might be less representative for local events, such as landslides*". If data are not representative the entire modelling is not representative.

***Thank you for voicing your concern. There are indeed limitations of a susceptibility analysis at 36-km resolution for specific hillslopes and we used some unfortunate phrasing to address the spatial representativeness issue. We removed the confusing statement and the text is updated as follows:*** *"Input uncertainty principally results from errors in the environmental data. To assess how input uncertainty propagates into the total prediction uncertainty, ensemble simulations can be used." (Line 58-61)*

***Furthermore, we will follow up on this and acknowledge the effect of spatial representativity and how that error is caught in the optimization of the "input" uncertainty. Please see our answer to comments 5 and 6 below. Yet, at this stage, it may already be relevant to point out that the native resolution of many of our input layers (e.g. slope) was much finer than 36 km and the variability within these large grid cells is (at least partially) taken into account. As explained in the introduction, the main reason for constructing a model at such a coarse resolution (and assess the associated uncertainties) is to make a***

*product that can be used in combination with currently existing soil moisture satellite observations (see below).*

4. Line 20 *LSS maps derived from environmental conditions are a fundamental tool for informing local population, city planners and decision makers both on the immanent landslide likelihood, but also about secondary effects such as major sediment sources* → this is true but very difficult to be applicable at the resolution of your analysis.

*We agree. This sentence was intended as a first introduction into the topic of landslide susceptibility. As we state in line 39-43, the susceptibility assessment in this study aims at a subsequent combination with satellite soil moisture products for spatio-temporal hazard assessment. This motivated the choice of a 36-km spatial resolution which is common for these satellite products. To prevent any misinterpretation, we will however update the sentence as follows:* "Regional high-resolution LSS maps derived from environmental conditions are a fundamental tool for informing local population, city planners and decision makers both on the immanent landslide likelihood, but also about secondary effects such as major sediment sources (Crozier, 2013; Maeset al., 2017; Broeckx et al., 2020). Large scale low-resolution LSS maps can serve as background information to be downscaled for the above applications at the local scale, or they can be used in conjunction with large-scale satellite data to construct a spatio-temporal estimate of the likelihood for a landslide" (Line 23-28)

*We would also like to point out that although gridded landslide susceptibility is usually assessed at finer resolution, the end result is still frequently aggregated into communal or provincial units which often cover areas comparable to one of our grid cells.*

5. (Line 60) The total uncertainty is estimated by comparing the predicted average LSS against the observed presence and absence of landslides → in your case the presence/absence of landslide is related to a too coarse grid resolution.

*Thank you for your remark. Indeed, the evaluation suffers from representativeness error, and that uncertainty is now included in the LSS uncertainty. The text is updated as follows:* "One such important source of uncertainty is spatial representativeness error (Blöschl and Sivapalan, 1995; van Leeuwen, 2015), especially when evaluating spatially averaged grid cell LSS estimates using single landslide observations as reference data." (Line 72-74)

6. *Due to their generalizing nature, LSS models are however prone to uncertainty.* → true but the uncertainty it is also highly related to the thematic data/landslide distribution/model used for the assessment. In your case the uncertainty associated to data and landslide distribution is more relevant than the entire modelling.

*We agree about these different sources of uncertainty in landslide modelling. This is why later on in the introduction (line 44 onwards) we introduce the concept of input and model uncertainty and stress the need for assessing the input uncertainty, especially at a 36-km resolution. We would nevertheless like to highlight that most of the environmental data, and especially the ones building on topography, originally come from much finer resolution (see also the answer to comment 10) and by aggregating the information, the error is – in fact – reduced at the coarse resolution (aggregation of noise, purely statistically). We will include the original spatial resolutions in Table 1 (see below).*

*Furthermore, the uncertainty of the exact landslide locations used for training of the LSS models (location accuracy ranging from 1 km to up to 25 km in the GLC) actually becomes less of a problem when aggregating the information into a 36-km grid cell.*

*Table 1 Predictor variables used in this study*

| Predictor variables | Data source | Original spatial resolution | Aggregation method to or within EASEv2, 36 km grid cell |
|---|---|---|---|
| slope (mean, maximum) [°] | USGS: details in Verdin et al. (2007) based on SRTM DEM[a] and GTOPO30[b] | 3" (SRTM DEM), 30" (GTOPO30) | mean and maximum |
| elevation (mean, standard deviation) [m a. s. l.] | CLSM parameters: details in Verdin (2013) based on SRTM DEM[a] and GMTED2010[c] | 3" (SRTM DEM), 7.5" (GMTED2010) | mean and standard deviation |
| depth to bedrock [m] | CLSM parameters: details in De Lannoy et al. (2014) based on GSWP-2[d] | 1° | spatial interpolation |
| percentage of gravel (0-30 cm) [vol%] | CLSM parameters details in De Lannoy et al. (2014) based on STATSGO2[e] and HWSD1.21[f] | 30" | most representative 30" sample |
| percentage of clay (0-30 cm and 0-100 cm) [w%] | | | |
| percentage of sand (0-30 cm and 0-100 cm) [w%] | | | |
| porosity (0-30 cm and 0-100 cm) [m³/m³] | | | |
| wilting point divided by porosity (0-30 cm and 0-100 cm) [-] | | | |
| compound topographic index, CTI (mean, maximum) = ln(specific catchment area/tan(slope)) [log(m)] | CLSM parameters: details in Verdin (2013) based on SRTM DEM[a] and GMTED2010[c] | 3" (SRTM DEM), 7.5" (GMTED2010) | mean and maximum |
| land fraction within grid cell | CLSM parameters: HYDRO1k based on GTOPO30, 1996 (EROS, 2018; Verdin, 2013) | 10" | areal fraction |
| fraction covered by each of 13 lithological classes [-]: *metamorphic rocks, mixed sedimentary rocks, siliclastic sedimentary rocks, basic plutonic rocks, acid plutonic rocks, basic volcanic rocks, intermediate volcanic rocks, carbonate sedimentary rocks, unconsolidated sediments, intermediate plutonic rocks, pyroclastics, evaporites, acid volcanic rocks* | GLiM created by Hartmann and Moosdorf (2012) | polygons | areal fraction |
| peak ground acceleration, PGA [m/s²] due to earthquakes expected with a return period of 475 years (i.e. 10% exceedance probability in 50 years) | GSHM[g] created by GSHAP[h] (Giardini et al., 2003) | 1° | nearest neighbour |
| rainfall climatological statistics [mm] | MERRA-2 (Bosilovich, 2015) | 0.625° lon x 0.5° lat | bilinear interpolation |
| surface soil moisture climatological statistics (0-5 cm) [m³/m³] | CLSM output | EASEv2, 36 km | - |
| root zone soil moisture climatological statistics (0-100 cm) [m³/m³] | | | |
| profile soil moisture climatological statistics (0-100 cm) [m³/m³] | | | |
| land surface temperature climatological statistics [K] | | | |
| runoff climatological statistics [mm] | | | |
| evaporation climatological statistics [mm] | | | |
| snow depth climatological statistics [mm] | | | |

7. (line 89) When you aggregate landslide data in a landslide location, do you check that your aggregation is reliable?

*Unfortunately, we do not fully understand this question. However, we have added some related text in response to other reviewer questions and hope this may help:*

*"Since LSS informs about the static environmental landslide likelihood, it is common practice to exclude the temporal aspect of landslide occurrence and instead work with landslide presence and absence locations. Multiple landslides within the same 36-km EASEv2 grid cell are therefore aggregated into one 'landslide presence grid cell', resulting in a total of $N_{LS}$=3757 (orange grid cells, Fig. A1). While we acknowledge that grid cells with more frequent landslide reporting can in general be expected to have a higher LSS, we found that the information about the frequency of landslide occurrence within a grid cell strongly mirrors biases in the landslide inventory, e.g. more landslides are reported in English-speaking countries. The aggregation, on the contrary, reduces the landslide presence reporting bias of the GLC." (Line 104-111)*

8. (Line 90) Multiple landslides within the same 36-km grid cell are aggregated into one 'landslide location --> The environmental condition selected in a 36 km grid cell can be completely inappropriate and not relevant to explain the landslide.

*Thank you for pointing this out. We are aware of the fact that a resolution of 36-km allows for rather large variations within the grid cell. This is caught in the ensemble uncertainty, because it implicitly accounts for spatial representativeness error. We would like to refer to the answer to comment 3 that this study's intention was to retrieve the likelihood of landslide occurrence for an area (of one grid cell) rather than single slopes.*

*Of course, one slope that is very prone for landslides can be situated in an area that is generally not susceptible to landslides. This is also visible in our results, where we do find low predicted LSS for some landslide presence grid cells (see e.g. grid cell 18 in Figure 5). We will include the following sentence in the discussion, section 5.2:* "At the same time, landslide presence grid cell 18 also has a very wide LSS distribution with a rather low average. This could either indicate that a non-hydrological process caused the landslide (misclassification) or that specific unrepresented features are present within the grid cell area." *(Line 402-404)*

9. (Line 111) *Absence grid cells are hence selected from grid cells 7 to 15 around a landslide occurrence* → How you can be sure that the selected conditions are not prone to landslides?

*Thank you for this remark. In our study, we follow the philosophy that "the past is the key to the future", as commonly done for landslide susceptibility approaches. This entails that for simplicity, we assume that areas where landslides have occurred in the past will also be prone to landslides in the future. Areas without historical landslide observations, are hence assumed to be not prone for landslides. This of course generally calls for a reliable landslide absence reporting, which for the GLC is only the case to a certain degree: "For large or remote areas, however, no reported landslide does not necessarily mean that the site never experienced [a landslide]." (Line 119-120) The introduction of the buffer and maximum radius was intended to tackle some of this issue by excluding grid cells in the vicinity of known landslide locations to be selected as a landslide absence grid cell. To be 100% sure of landslide absence, we would need to be in the field or do intensive visual assessments based on google Earth as has for example been done by Depicker et al. (2021). The approach taken in this study can be regarded as our best educated guess.*

*We will update the text as follows to reflect better on the issue:* "[…] it is still possible that an absence grid cell could experience a landslide, even if none has been reported in the GLC. A prominent example of this are absence grid cells 1 and 7, located in the East African Rift and India, respectively. Both grid cells have no reported landslide, but very wide LSS distributions, with relatively high LSS values. This discrepancy between prediction and observation could indicate the need to visit this location for landslide research. […] Overall, we find an average $\overline{LSS}_{2500}$ of 0.18 (0.82) for landslide absence (presence) grid cells (as displayed in Fig. A1) which makes us confident in our classification of these grid cells. "(Line 398-405)

10. To compute the compound topographic index, you need the specific catchment area and the slope. How do you measure then in a 36 km grid?

*Indeed, the CTI was originally calculated per catchment based on 3'' data from SRTM observation south of 60°N, and on 30'' data from GTOPO30 for the high northern latitudes (as described in Verdin 2013, see Table 1). The values that we use are the mean and maximum CTI per 36-km grid box (see Table 1). We have now added the resolution of the datasets that were used to generate CLSM model parameters in Table 1 (see comment 6).*

11. Why do you consider the peak ground acceleration if you want to evaluate hydrologically triggered landslides?

*Thank you for this question. While local lithology is an essential predictor for landslide susceptibility assessment, various studies have shown that lithological classes alone tell only part of the story. Even when accounting for topography and lithology, seismic proxies like PGA play a key role in explaining regional, continental and global patterns of landsliding. This is also the case across regions where seismicity is overall too weak to directly trigger landslides (e.g. Vanmaercke et al., 2017; Broeckx et al., 2018; Stanley et al., 2021). The most likely reason for this is that weak yet prolonged seismic and tectonic activity can have large effects on rock fracturation and, by extent, weathering and lithological strength (e.g. see discussion by Molnar et al., 2007). As such, peak ground acceleration can be a highly relevant preparatory factor that explains landslide susceptibility, even for hydrologically triggered landslides. We will extend this explanation to prevent confusion on this aspect:* *"Peak ground acceleration (PGA) is the likely level of ground motion from earthquake (Giardini et al., 2003). Here, we do not use it as the likelihood of a seismic landslide trigger, but rather as a proxy for the fracturation and weakening that lithologies have undergone due to seismic and tectonic activity (Lin et al., 2017; Vanmaercke et al., 2017; Broeckx et al., 2018)" (Line 156-160)*

12. (Line 161) → The mixed effects approach allows us to include a so-called 'random effect', here the random intercept α, for which we use the average road network density stratified into 6 groups (divided by the global quintile thresholds) → not clear

*Thanks for pointing this out. For the logistic regression in this study, we decided to have the intercept vary with stratified road network density in order to prevent this bias in the landslide reporting from affecting the retrieved connection to the environmental predictors. We will update the text as follows:* *"The mixed effects approach allows us to include a categorically scaled variable as a so-called 'random effect', here the random intercept α, for which we use the average road network density (RND) stratified into 6 classes. We summarize all land grid cells where average RND is negligible (< 1 m/km²) into the first class and use quantiles 20, 40, 60 and 80 of those grid cells with non-negligible RND to divide the rest into additional 5 classes. The mixed effects approach will then result in one global logistic regression equation that has the same $\beta$-factors for all grid cells, but different α values according to each grid cell's RND class. The 6 α values are assumed to come from a zero-mean normal distribution (Zuur, 2009)." (Line 184-190)*

13. (Line 185) We group the grid cells into a total of 100 blocks according to climatological conditions within 10 predefined regions (roughly two per continent), independent of landslide absence or presence → a) this means that each block has about 75 pixel?

*This is approximately correct. Since we do not enforce each block to consist of the same number of grid cells (pixels), the number of grid cells per block varies, with the median being 55. We will mention this as part of the alterations in answer to comment 14.*

b) What is the rational to select roughly two climatological conditions per continent? If this is not a consistent selection is not representative.

*Thank you for your concern. The described method does not aim at concrete climatologic classification but rather at an appropriate process to delineate the aforementioned "blocks". The (sub-)continents were delineated based on our expert opinion, with the intention of a very broad first climatological stratification: North America west, North America east, South America west, South America east, Europe, Africa west, Africa east, Asia east, Asia west, Australia-Oceania. Within those (sub-) continents, the kmeans clustering according to average climatological conditions divided grid cells with similar climatological regimes into 10 groups that we refer to as "blocks". We assume the climatological regimes to also be representative of the landslide regimes. The so grouped grid cells of*

*one block do not necessarily have to be neighboring. (Please see our answer to comment 14 for some additional details.)*

*This grouping varies per CV model creation (with changing subsets used for training and testing), but we don't see this as an issue. Had we opted for a random selection of pixels, which is the most common approach in LSS modelling, there would also not have been any consistency in the assigning to different subsets.*

*We will add the following sentence:* *"Note that the definition of the individual blocks varies between each repetition of absence grid cell sampling due to the kmeans clustering algorithm." (Line 233-234)*

14. (Line 183) One subset consists of 20 randomly sampled 'blocks', i.e. small groups, of the 7514 grid cells selected for model creation. We group the grid cells into a total of 100 blocks according to climatological conditions within 10 predefined regions (roughly two per continent), independent of landslide absence or presence. Within these regions, we mimic typical climatological zonations (for example that of Köppen) through k-means clustering (Lloyd, 1982) of 30-year average soil surface temperature and rainfall (see Table 1), dividing each region into 10 blocks. → Not clear the relation between the blocks and the 5 subset.

*Thank you for this remark. In line with our reply to comment 13, we will alter the text as follows:* *"We employ a blocked random CV (B-CV), as recommended by Roberts et al. (2017), which we found to indeed yield most realistic error estimates in comparison to random or spatial sampling (not shown). This means that instead of randomly sampling individual grid cells into the 5 subsets for training and testing the model as part of CV, we randomly sample small groups of grid cells with similar environmental conditions, so-called "blocks" (see Fig. 1). We expect that the environmental conditions are similar in neighboring pixels (for example same subcontinent) and for similar climate zones. We therefore derive blocks in 2 steps. First, the 7514 grid cells selected for model creation are divided according to 10 predefined (sub-) continents. Within each (sub-) continent, we then derive in a second step 10 blocks through kmeans clustering (Lloyd, 1982) of 30-year average soil surface temperature and rainfall (see Table 1). In total we retrieve 100 blocks comprising different numbers of grid cells (median: 55) that are not necessarily located next to each other. The 100 blocks are then randomly divided into the 5 subsets for model creation (20 each)." (Line 215-228)*

*A final distribution of the 5 subsets might then look like this:*

[Figure]

*Figure 1 Spatial distribution of landslide presence and absence grid cells selected for LSS model creation. Colors indicate the subset they were sampled into, based on blocked random selection. One point on the map indicates the center of a grid cell.*

15. (Line 224) Aggregated data vs observations → Do you really think is reliable to aggregate original observations? In Italy for example you have aggregated 5438 observations in 309 points. I think the two data are completely different and infact you get very low ROC curve.

*Indeed, the aggregation has a quite extreme effect for Italy as compared to the other validation data sets of Russia and Africa. Please note, however, that the aggregation transforms the original landslide observation points into landslide presence grid cells. Essentially, what we learn is that in most areas of Italy, you can find landslides and should expect a high landslide susceptibility. The reason for the low AUC value was actually introduced by a mistake on our side that we now discovered: For the ROC analysis, we had selected a box around the reference validation data set region rather than really cutting at the country or continental borders. For Italy that meant that a large number of very high susceptibility grid cells in the Alps joined the analysis as false positives, while we actually did not have validation data for these grid cells. This mistake has been corrected, and we now obtain great AUC values of 0.91 for Italy, 0.84 for Africa and 0.92 for Russia.*

[Figure]

*Figure 2 Corrected ROC curves and AUC values for the validation landslide inventories, see Figures 6 and 7 in manuscript*

*In addition, we will add the following thought in the text: "Future research could explore the additional information, such as landslide sizes, types or the frequency of occurrence per grid cell instead of reducing the data to landslide presence and absence. For the latter, one would need to find ways to counteract the English-language and economic bias of the GLC which is more pronounced when using the actual number of reports instead of the presence-absence method chosen in this study."(Line 452-456)*

16. (Line 236) Values of the intercept, which is part of all models, vary with road network density as part of the MELR and mostly have and average close to zero (not shown) → can you explain better this statement?

*Thank you for your question. Hopefully, we were already able to provide some clarification concerning the varying intercepts in MELR in answer to comment 12 . For each average road network density $(\overline{RND})$ group, we retrieve one intercept ($\alpha$). These are both positive and negative, i.e. increase or decrease the susceptibility resulting from the predictor variables depending on $\overline{RND}$ within the grid cell. That they have an average of zero is actually an initial condition, which we mention now already in section 3.1 (please see our answer to comment 12 for more details): "The 6 $\alpha$ values are assumed to come from a zero-mean normal distribution (Zuur, 2009)." (Line 190)*

*For low (high) $\overline{RND}$, $\alpha$ takes negative (positive) values. This means we move the connection between $\overline{RND}$ and landslide presence into the intercept instead of using $\overline{RND}$ as a predictor variable itself. We will alter the sentence as follows:* "The values of the intercept α take negative values for low RND and positive values for high RND (by design, not shown)." (Line 294)

17. Fig. 3 → how much all this complex analysis improves/enhances at worldwide scale a simple regression model applied to obtain an LLS?

*Thank you for this question. For individual grid cells where the accuracy was low, i.e. where the difference between predicted LSS and observed absence (0) or presence (1) is large, the ensemble approach was able to improve the accuracy in comparison to the CV ensemble (see answer to comment 3 from reviewer #1). If the interest lies only in average LSS assessment (without uncertainty) to retrieve information on the general global patterns, it is not much of a drawback to use a simple regression model.*

*We will expand the changes made in answer to comment 4 from reviewer #1:* "The AUC values of ensemble averages remain practically the same, and an LSS model without predictor perturbations would hence suffice for a general insight in the global spatial LSS pattern." (Line 429-431)

*The key advantage of this study is that we obtain uncertainty estimates that can be used in conjunction with satellite data in a Bayesian framework, which is now better clarified in the text:* "A reliable uncertainty assessment of global LSS estimates is moreover crucial when subsequently combining them in a statistically optimal way with, for example, satellite soil moisture products from Soil Moisture Ocean Salinity (SMOS) or Soil Moisture Active Passive (SMAP) as used by Felsberg et al. (2021)." (Line 39-43)

18. (Line 265) The LSS2500 map hence performs very well over Russia and Africa, while showing some difficulties to capture the patterns for Italy→ This is the situation where you have modelled 309 points aggregated from 5438 observation. Are you sure that the aggregated points are representative of the failure distribution around the world?

*In answer to this, we would like to point to the answers of comments 3, 8, and 15. The aggregation of 5438 observation points into 309 landslide presence grid cells of 36-km resolution is reasonable when interested in an area's likelihood of landslides. We are convinced that the aggregated landslide presence grid cells for Africa, Russia and Italy paint a realistic pattern of areas that are prone to landslides within the continent or country.*

*Concerning the failure distribution around the world, Figure A.1 shows the 3757 landslide presence grid cells as aggregated from 12515 landslides reported in the GLC (see lines 103-108). The landslide presence grid cells cover all prominent landslide hot spots, as for example also visible in the original GLC point observations (Kirschbaum et al. 2015) or found by Froude and Petley (2018).*

19. The procedure is quite complex and the real meaning between the LSS2500 and LSS100 is not very easy to understand

*The $LSS_{100}$ is an ensemble of 100 LSS maps that is the direct result of MELR with blocked random cross validation (5 subsets) and 20 times repetition of the absence grid cell subsampling. In addition to that, $LSS_{2500}$ includes predictor variable perturbations, so that per 1 map in $LSS_{100}$, you have 25 maps in $LSS_{2500}$. This is introduced in lines 229-238:*
"[…] the MELR […] results in 5 different model equations and corresponding LSS maps. By repeating the absence sampling 20 times, we obtain a total of 100 LSS maps (referred to as CV ensemble or $LSS_{100}$, see Fig. 1)[…]"

*For the input ensemble perturbations, we apply one fitted model equation to a slightly perturbed set of its predictor variable values. In total, 25 repetitions of this process are conducted […], this results in a total amount of 2500 LSS maps (referred to as full ensemble or $LSS_{2500}$) […]"*

**We will nevertheless update the text in a later part of the manuscript to be clearer about both types of ensembles in section 4.3:** *"The above discussion of the full ensemble $LSS_{2500}$ includes perturbations to the predictor variables on top of the CV ensemble $LSS_{100}$ obtained by the CV techniques alone." (Line 333-335)*

---

## Referee Report (RR1)

The paper has in my opinion been significantly improved. But I also think the general approach taken by the authors is still not clearly described, and that improvement must still be done in the presentation of the paper.

My main comments are as follows. Again, I stress that I have no expertise whatsoever on the occurrence of landslides, and my comments bear exclusively on the methodological aspects and the presentation of the paper.

The line numbers below refer to file nhess-2021-360-ATC1.pdf, *i.e.* the file containing the Author's Tracked Changes.

1. Although I think I have now basically understood what the authors have done, I still think the paper would be difficult to understand for a non-expert reader. For instance, I not found that Figure 1, which is meant to describe the general methodological approach of the paper, is really useful.

1*a*. The general methodological approach is described in Section 3, and particularly in subsection 3.2, entitled *Cross validation (CV) and input perturbations for reliable uncertainty estimation*. The subsection begins with introduction of the Brier score (Eq. 2). From what I understand, the Brier score has nothing to do with either cross validation or input perturbations *per se*, but only with the assessment of the input perturbations. It should be introduced at a later stage, and certainly after cross validation has been introduced.

1*b*. In any case, a 'predicted average' $LSS^{bar}$ (sorry, no upper bar on my text editor) is introduced on the occasion of the Brier score. It is not clearly said which kind of average that is, nor on which kind of prediction it is obtained. By the time the Brier score is introduced, it should have already been said that the output of the entire estimation process essentially consists of two $LSS$ maps, *viz.* $LSS_{100}$ and $LSS_{2500}$, and that $LSS^{bar}$ will normally be, for each grid cell $i$, the average of $LSS$ over one of those two maps.

1*c*. Then, just after Eq. (2) (l. 217), reference is made to an undefined ensemble variance $\sigma^2_{LSS}$ which has nothing to do with the Brier score, and to an undefined *'actual'* *uncertainty*.

1*d*. It is only later, in the course of rather intricate explanations, that the two maps $LSS_{100}$ and $LSS_{2500}$ are introduced (ll. 237-238 for $LSS_{100}$, and ll. 244-245 for $LSS_{2500}$).

All that is only to stress how confusing the paper can be for a reader who is an outsider. I suggest that, as is a common practice in scientific literature, the authors end their Introduction with brief description of the text that will follow, with what will be the content of each Section.

And, for another example, the authors write (ll. 236-237) *This results in 5 different model equations* ... Simply writing *model equations of form (1)* ... would make it much easier for the reader.

2. Ll. 137-138, *We* [...] *sample from the absence grid cells* ... Random sampling, or what ?

Then, later on (l. 237), *By repeating the absence sampling 20 times* .... Is that a new sampling of the same kind as in ll. 137-138, or something else ? Please clarify.

L. 255, … *the logistic regression* […] *is asymptotic*. What do you mean by *asymptotic* (you use the same word on a number of other occasions, for instance ll. 433-435) ?

Figures 2 (right) and 7. The exact meaning of boxes and vertical lines (total spread ?) does not seem to be mentioned.

Ll. 195-196, *The 6 α-values are assumed to come from a zero-mean normal distribution*. What does that mean ? The text that follows says that the quantity RND has not been used as a predictor variable, but does not really explain how the parameter $\alpha$ has been defined.

Ll. 177-178, *A one unit change in the predictor variable $x_i$ results in a muliplicative change in the odds of landslide presence by exp($\beta_i$)*. Well, only for small $P(Y=1)$

Figure 4b does not seem to be commented upon. There is no point in including a figure in a paper if it not for saying what conclusion, however succinct, must be drawn from it.

Ll. 261-262, *true positive rate ,false positive rate*. Explain

Ll. 324 and 409, *Sinai peninsula,* actually the Sinai peninsula is a very small region at a global scale. I suspect you mean *Arabian peninsula*
Same lines, *Sahara → a large part of Africa*

Ll. 347-348, … *the ensemble averages* […] *are similar, ….* That is actually visible from the bottom panel 4c. Correct accordingly.

L. 433, $LSS_{2500}$- $\sigma_{LSS2500}$ (not $LSS_{2500-\sigma LSS2500}$)

---

## Author Response (AR2)

The authors thank the reviewer for the constructive comments. The comments are shown in regular fonts, *our responses are in bold italic, blue fonts*. *Changes made in the manuscript are printed in italic, underlined, blue fonts. Our line references refer to the updated manuscript with track-changes.*

*In addition to the answers below, we implemented a few minor editorial changes that are marked in the track changes.*

| Reviewer #1 |
| --- |

The paper has in my opinion been significantly improved. But I also think the general approach taken by the authors is still not clearly described, and that improvement must still be done in the presentation of the paper.

My main comments are as follows. Again, I stress that I have no expertise whatsoever on the occurrence of landslides, and my comments bear exclusively on the methodological aspects and the presentation of the paper.

*We thank the reviewer for the detailed feedback on the methods that help to improve the readability of the paper and understanding of our research even further.*

The line numbers below refer to file nhess-2021-360-ATC1.pdf, *i.e.* the file containing the Author's Tracked Changes.

1. Although I think I have now basically understood what the authors have done, I still think the paper would be difficult to understand for a non-expert reader. For instance, I not found that Figure 1, which is meant to describe the general methodological approach of the paper, is really useful.

*Thank you for this feedback. We have now revised the figure to only show the essential aspects of the approach followed in this study:*

[Figure]

*Figure 1  Schematic of methodology used in this study to derive ensembles of global landslide susceptibility (LSS) maps. 'Ensemble' refers to a collection of LSS maps. In the course of this study, we refer to different subsets of the **full ensemble** ($LSS_{2500}$), namely the ensemble from one single blocked random CV application (**single CV ensemble**, $LSS_5$), when adding input perturbations to it (**partial ensemble**, $LSS_{125}$) or when repeating the underlying landslide absence subsampling (**CV ensemble**, $LSS_{100}$). Subscript numbers indicate the size of the LSS ensemble. Model fitting performance is evaluated during the process of CV by calculating the area under the Receiver Operating Characteristic curve (AUC) for each model equation.*

1a. The general methodological approach is described in Section 3, and particularly in subsection 3.2, entitled *Cross validation (CV) and input perturbations for reliable uncertainty estimation*. The subsection begins with introduction of the Brier score (Eq. 2). From what I understand, the Brier score has nothing to do with either cross validation or input perturbations per se, but only with the assessment of the input perturbations. It should be introduced at a later stage, and certainly after cross validation has been introduced.

***Thank you for this feedback. To increase readability, we moved the introduction of the Brier score to later in this section and section 3.2. starts as follows:***

*"In this study, the predicted total ensemble uncertainty results from the combination of CV techniques and input ensemble perturbations." Lines 218-219*

1b. In any case, a 'predicted average' $LSS^{bar}$ (sorry, no upper bar on my text editor) is introduced on the occasion of the Brier score. It is not clearly said which kind of average that is, nor on which kind of prediction it is obtained. By the time the Brier score is introduced, it should have already been said that the output of the entire estimation process essentially consists of two $LSS$ maps, viz. $LSS_{100}$ and $LSS_{2500}$, and that $LSS^{bar}$ will normally be, for each grid cell $i$, the average of $LSS$ over one of those two maps.

*Thank you for this feedback. $\overline{LSS}$ is the average of all LSS values (100 or 2500) per grid cell, and the Brier Score is the sum over all grid cells of the difference between $\overline{LSS}$ and the corresponding observation (landslide presence = 1, landslide absence = 0). We added "per grid cell" to the introduction of the averages to make its definition clearer:*

*"[...] we obtain a total of 100 LSS maps [...] that allow for calculations of an ensemble average LSS ($\overline{LSS}_{100}$), as well as a standard deviation ($\sigma_{LSS_{100}}$) per grid cell." Lines 233-234*

*"In combination with the 5 model equations and 20 repetitions for the CV ensemble, this results in a total amount of 2500 LSS maps [...] with corresponding average ($\overline{LSS}_{2500}$) and standard deviation ($\sigma_{LSS_{2500}}$) per grid cell." Lines 238-240*

*With the feedback from 1.b, we have moved the introduction of the Brier Score to after these averages are introduced.*

1c. Then, just after Eq. (2) (l. 217), reference is made to an undefined ensemble variance $\sigma^2 LSS$ which has nothing to do with the Brier score, and to an undefined '*actual*' uncertainty.

*With the new order, the variance is now introduced after the ensemble standard deviations ($\sigma_{LSS_{100}}$ and $\sigma_{LSS_{2500}}$), so that it should be understandable. We moreover now clearly introduce what the phrase 'actual' uncertainty refers to:*

*"The aim is to design an LSS model setup so that the predicted total ensemble uncertainty, quantified by the ensemble variance or spread ($\sigma_{LSS}^2$) matches the discrepancy between predictions and observations which we refer to as the 'actual' uncertainty. A measure of this actual uncertainty is the Brier Score (BS) (Wilks, 2011) which compares the predicted average LSS ($\overline{LSS}$) against landslide observations from the GLC (o) at different grid cells i (i = 1; ...;N): [...]*

*This actual uncertainty by design includes model and input error ($\overline{LSS}$), but also error in the reference data (o), and spatial representativeness error." Lines 241-247*

1d. It is only later, in the course of rather intricate explanations, that the two maps $LSS_{100}$ and $LSS_{2500}$ are introduced (ll. 237-238 for $LSS_{100}$, and ll. 244-245 for $LSS_{2500}$).

All that is only to stress how confusing the paper can be for a reader who is an outsider. I suggest that, as is a common practice in scientific literature, the authors end their Introduction with brief description of the text that will follow, with what will be the content of each Section.

*We added a paragraph providing an overview at the end of the introduction, and additionally at the beginning of Section 3. We hope that this helps to avoid confusion of the reader.*

*"Section 2 introduces the landslide (presence, absence) and environmental data used to create ensemble LSS maps. The LSS model construction based on MELR is introduced in Sect. 3, along with the methods of CV and input predictor variable perturbations for uncertainty assessment, and methods to evaluate the results. Section 4 presents the resulting LSS model structure and selected predictor variables, and the ensemble LSS evaluation for different input perturbations. Section 5 discusses various aspects of the results. The paper closes with a summarizing conclusion." Lines 86-90*

*"This section introduces the methods used in this study for model construction and evaluation. Section 3.1 introduces the general principles of logistic regression used to derive global LSS estimates, before elaborating the predictor variable selection process and the implementation of average road network density as a random effect. Section 3.2 introduces methods for uncertainty assessment. First, cross validation is introduced with a detailed explanation of the blocked random sampling. Second, the methods of input ensemble perturbations are briefly explained (details are elaborated in Appendix A2). LSS results based on the first approach alone are referred to as 'CV ensemble' or $LSS_{100}$. Results based on both CV and input ensemble perturbations are referred to as 'full ensemble' or $LSS_{2500}$. Section 3.3 introduces the methods and data used for the evaluation of ensemble average LSS and the impact of the extended uncertainty assessment through input perturbations."* Lines 159-166*

And, for another example, the authors write (ll. 236-237) *This results in 5 different model equations …* Simply writing *model equations of form (1) …* would make it much easier for the reader.

***Thank you for this suggestion. We added it to the manuscript:***

*"This results in 5 different model equations of form Equation 1 and corresponding LSS maps."* Lines 231-232*

2. Ll. 137-138, *We […] sample from the absence grid cells …* Random sampling, or what ?

Then, later on (l. 237), *By repeating the absence sampling 20 times ….* Is that a new sampling of the same kind as in ll. 137-138, or something else ? Please clarify.

***Yes, the subset of landslide absence grid cells is randomly sampled and this random subsampling is repeated. We added :***

*"We therefore randomly sample from the absence grid cells […]"* Lines 132-133*

*"By repeating the random absence grid cell sub-sampling 20 times, we obtain a total of 100 LSS maps […]"Lines 232-233*

3. L. 255, *… the logistic regression […] is asymptotic.* What do you mean by *asymptotic* (you use the same word on a number of other occasions, for instance ll. 433-435) ?

***By "asymptotic" we want to stress that the logistic regression function is bound to the interval (0,1) following an S-shape: For intermediate predictor variable values, P(Y=1) behaves quasi-linearly. At the upper or lower edge of the predictor variable spaces, however, variations in the exponent ($\alpha + \sum_{i=1}^{n} \beta_i \cdot x_i$) do not propagate into P(Y=1). We added a sentence to clarify this at the first mention of the phrase "asymptotic" in Section 2.3:***

*"Note that these perturbations in $x_j$ do not linearly propagate into the LSS estimates, because the logistic regression (see Equation 1) relates $x_j$ to LSS via an S-shape curve, with quasi-linear behaviour at the center (i.e. intermediate $x_j$ values) and asymptotic behaviour towards the upper or lower limit (i.e. for very low or high $x_j$ values). Locations of largest perturbation do thus not necessarily coincide with large resulting ensemble uncertainty."* Lines 254-257*

4. Figures 2 (right) and 7. The exact meaning of boxes and vertical lines (total spread ?) does not seem to be mentioned.

***Figures 2 (right) and 7 used Tukey's definition of boxplots, where boxes describe the inter-quartile range (IQR), i.e. from Quantile 25 (Q25) to Quantile 75 (Q75), and whiskers extend from the minimum within Q25-1.5*IQR to the maximum within Q75+1.5*IQR. Everything outside this range is defined as outlier (not shown in Figures 2 and 7). To avoid unnecessarily complicated explanation in the paper, we updated Figures 2 and 7 with whiskers now extending from the overall minimum to maximum and***

*indicate this definition in the captions as well as upon the first mention of box plots in the text (see below). We also updated Figure 8c-d where whiskers formerly extended from the 5th to 95th quantile to have all boxplots shown in the paper of the same definition. The conclusions from the data remain unchanged by these changes in the visualization.*

*"The right panel shows boxplots of the $\beta$-values for each predictor variable (see Equation 1). Whiskers extend from minimum to maximum and boxes from 25th to 75th quantile, with the median indicated in between." Lines 295-296*

[Figure]

*Figure 2 (Left) Frequency of selected predictor variables and (Right) corresponding $\beta$-values. [...] Whiskers extend from minimum to maximum $\beta$-values. [...]*

[Figure]

*Figure 3 Distribution of AUC for model fitting performance (test data) and model prediction performance [...]. Boxplots are shown for CV ensemble members ($LSS_{100}$) and full ensemble members ($LSS_{2500}$, including CV ensemble members), with whiskers extending from minimum to maximum AUC. [...]*

[Figure]

*Figure 4 Comparison of $\overline{LSS}_{2500}$ against existing global categorical LSS maps […]. Boxplots show distributions of $\overline{LSS}_{2500}$ values extracted from the nearest 36-km grid cell for each (c) 1-km and (d) 0.5° grid cell in the reference map per LSS class. Whiskers extend from minimum to maximum $\overline{LSS}_{2500}$. […]*

5. Ll. 195-196, *The 6 α-values are assumed to come from a zero-mean normal distribution*. What does that mean ? The text that follows says that the quantity RND has not been used as a predictor variable, but does not really explain how the parameter α has been defined.

*Thank you for this question. The parameter α, being the intercept in the exponent of the logistic regression function (1), is obtained during the model fitting process. For traditional logistic regression (fixed effect), this would be one parameter value for the whole training data set, fit alongside the β-factors by maximum likelihood estimation. For the mixed effects logistic regression, the best fitting $\alpha_j$ values are obtained separately for the j=6 RND groups, while ensuring that the β-values remain the same for the whole training data set, i.e. across all RND groups. In the model fitting algorithm glmer in R, the variation of these $\alpha_j$-values around an "average" intercept or α, is defined to be from a zero-mean normal distribution (see Zuur et al. 2009).*

*We acknowledge that the phrasing was unfortunate, and altered the manuscript as follows:*

*"The mixed effects approach will then result in one global logistic regression equation that has the same β-factors for all grid cells, but 6 different α-values according to each grid cell's RND class. For model fitting purposes it is assumed that these 6 α-values come from a normal distribution (Zuur, 2009)." Lines 194-197*

6. Ll. 177-178, *A one unit change in the predictor variable $x_i$ results in a multiplicative change in the odds of landslide presence by exp(βᵢ).* Well, only for small P(Y=1)

*Thank you for this remark. The multiplicative change we refer to is in the odds, i.e. the ratio of P(Y=1) and P(Y≠1) = 1-P(Y=1), and not in P(Y=1) itself. More specifically, the multiplicative change is obtained*

*by comparing the odds for $x_i+1$ to those of $x_i$. This so-called odds ratio (Zuur et al. 2009) simplifies to $exp(\beta_i)$, which is the multiplicative change we refer to.*

$$odds = \frac{P(Y=1)}{1-P(Y=1)} = exp(\alpha + \sum_{i=1}^{n} \beta_i \cdot x_i)$$

$$odds\ ratio = \frac{odds(x_i+1)}{odds(x_i)} = exp(\beta_i)$$

*While not linearly connected, an increase in the odds goes along with an increase in LSS (as illustrated on the figures below). We added a sentence explaining the concept of the odds and the connection with LSS:*

*"A one unit change in the predictor variable $x_i$ results in a multiplicative change by exp($\beta_i$) in the odds of landslide presence, defined as the ratio of P(Y = 1)/(1-P(Y = 1)) = $exp(\alpha + \sum_{i=1}^{n} \beta_i \cdot x_i)$. An increase in the odds of landslide presence is associated with a (non-linear) increase in LSS. Positive (negative) $\beta$-values hence indicate an increase (decrease) in LSS with an increase in the predictor variable." Lines 176-179*

[Figure]

*Figure 5 Illustration of the connection between LSS and the odds*

7. Figure 4b does not seem to be commented upon. There is no point in including a figure in a paper if it not for saying what conclusion, however succinct, must be drawn from it.

*Thank you for this remark. We updated the discussion of Figure 4 as follows (see more in response to comment #10 below):*

*"Figure 4a and b show that the LSS uncertainty is a function of the average LSS values[…]" Line 329-330*

8. Ll. 261-262, *true positive rate , false positive rate.* Explain

*We added an explanation of these two in the manuscript:*

*"For the ROC, the true positive rate of one LSS map is displayed against its false positive rate for different possible thresholds in the continuous probability (here: LSS) that is predicted. The true positive rate is the proportion of correctly predicted landslide presence grid cells when applying said threshold ('true positives') of all observed landslide presence grid cells (Wilks, 2011). The false positive rate is the*

*proportion of erroneously predicted landslide presence grid cells ('false positives') of all observed landslide absence grid cells." Lines 261-265*

9. Ll. 324 and 409, *Sinai peninsula*, actually the Sinai peninsula is a very small region at a global scale. I suspect you mean *Arabian peninsula* Same lines, *Sahara → a large part of Africa*

***Thank you for this remark. We updated these in the manuscript.***

*"Very flat areas or planes, such as central northern Canada, Siberia, the Tibetan plateau, the Arabian peninsula, large parts of Africa (especially the Sahara) as well as central Australia have very low $\overline{LSS}_{2500}$." Lines 313-315*

*"At the same time, $\overline{LSS}_{2500}$ shows much less variation than the map by Stanley and Kirschbaum (2017) within large deserts (Sahara, Arabian peninsula and central Australia)."Lines 396-397*

10. Ll. 347-348, *… the ensemble averages […] are similar*, …. That is actually visible from the bottom panel 4c. Correct accordingly.

***We agree and revised the discussion of Figure 4 in the manuscript as follows:***

*"Figure 4a and b show that the LSS uncertainty is a function of the average LSS values and that $\sigma_{LSS_{2500}}$ is typically higher than $\sigma_{LSS_{100}}$. Figure 4d shows that the differences between $\sigma_{LSS_{2500}}$ and $\sigma_{LSS_{100}}$ are smallest for the very high and low $\sigma_{LSS_{100}}$. However, Figure 4c shows that the ensemble averages $\overline{LSS}_{2500}$ and $\overline{LSS}_{100}$ are similar, as expected from the additional zero-mean predictor variable perturbation. The values of $\overline{LSS}_{2500}$ are slightly smaller than those of $\overline{LSS}_{100}$, except for very small $\overline{LSS}$ (<0.1). " Lines 329-335*

11. L. 433, $LSS_{2500}$- $\sigma_{LSS2500}$ (not $LSS_{2500}$- $_{\sigma LSS2500}$)

***Thank you for spotting this. This is an unintended artifact of the pdf rendering, and to avoid this, we rephrase the text as follows:***

*"The reasons for this relationship between $\overline{LSS}_{2500}$ and $\sigma_{LSS_{2500}}$ are twofold: […]" Lines 419*